# Pharmacological inactivation of the prion protein by targeting a folding intermediate

Giovanni Spagnolli [1,2,13], Tania Massignan[1,2,3,13], Andrea Astolfi[4,13], Silvia Biggi[1,2], Marta Rigoli [5], Paolo Brunelli[1,2], Michela Libergoli[1,2], Alan Ianeselli[1,2], Simone Orioli[5,6], Alberto Boldrini [1,3], Luca Terruzzi [1,3], Valerio Bonaldo [1,2], Giulia Maietta[1,2], Nuria L. Lorenzo[7], Leticia C. Fernandez[7], Yaiza B. Codeseira[7], Laura Tosatto[8], Luise Linsenmeier[9], Beatrice Vignoli[5], Gianluca Petris [1], Dino Gasparotto[1,2], Maria Pennuto [10,11], Graziano Guella [5], Marco Canossa[1], Hermann C. Altmeppen [9], Graziano Lolli[1], Stefano Biressi [1,2], Manuel M. Pastor[12], Jesús R. Requena[7], Ines Mancini[5], Maria L. Barreca [4✉], Pietro Faccioli[5,6✉] & Emiliano Biasini [1,2✉]

Recent computational advancements in the simulation of biochemical processes allow investigating the mechanisms involved in protein regulation with realistic physics-based models, at an atomistic level of resolution. These techniques allowed us to design a drug discovery approach, named Pharmacological Protein Inactivation by Folding Intermediate Targeting (PPI-FIT), based on the rationale of negatively regulating protein levels by targeting folding intermediates. Here, PPI-FIT was tested for the first time on the cellular prion protein (PrP), a cell surface glycoprotein playing a key role in fatal and transmissible neurodegenerative pathologies known as prion diseases. We predicted the all-atom structure of an intermediate appearing along the folding pathway of PrP and identified four different small molecule ligands for this conformer, all capable of selectively lowering the load of the protein by promoting its degradation. Our data support the notion that the level of target proteins could be modulated by acting on their folding pathways, implying a previously unappreciated role for folding intermediates in the biological regulation of protein expression.

[1] Department of Cellular, Computational and Integrative Biology, University of Trento, 38123 Povo, TN, Italy. [2] Dulbecco Telethon Institute, University of Trento, 38123 Povo, TN, Italy. [3] Sibylla Biotech SRL, 37121 Verona, VR, Italy. [4] Department of Pharmaceutical Sciences, University of Perugia, 06123 Perugia, PG, Italy. [5] Department of Physics, University of Trento, Povo, Trento, TN, Italy. [6] INFN-TIFPA, University of Trento, Povo, Trento, TN, Italy. [7] CIMUS Biomedical Research Institute, University of Santiago de Compostela-IDIS, Santiago de Compostela, Spain. [8] Institute of Biophysics, National Council of Research, 38123 Povo, Trento, TN, Italy. [9] Institute of Neuropathology, University Medical Center Hamburg-Eppendorf, 20246 Hamburg, Germany. [10] Department of Biomedical Sciences (DBS), University of Padova, 35131 Padova, Italy. [11] Veneto Institute of Molecular Medicine (VIMM), 35129 Padova, Italy. [12] RIAIDT, University of Santiago de Compostela-IDIS, Santiago de Compostela, Spain. [13] These authors contributed equally: Giovanni Spagnolli, Tania Massignan, Andrea Astolfi. ✉email: maria.barreca@unipg.it; pietro.faccioli@unitn.it; emiliano.biasini@unitn.it

Protein expression and function in eukaryotic cells are tightly harmonized processes modulated by the combination of different layers of regulation. These may occur at the level of gene transcription, processing, stability, and translation of the mRNA as well as by assembly, post-translational modifications, sorting, recycling, and degradation of the corresponding polypeptide[1]. In addition, the expression of a small subset of proteins is known to be regulated by the presence of specific ligands, which are required along the folding pathway to reach the native, functional state[2]. Typical examples include nuclear receptors, such as estrogen and androgen receptors[3]. The integration between these pathways and the protein quality control machinery, deputed to avoid the production and accumulation of aberrantly folded proteins, is known as proteostasis[1,4]. Mechanisms by which proteostasis is ensured include chaperone-assisted protein folding as well as re-routing and degradation of misfolded protein conformers. Consistently, a large percentage of human pathologies are linked to alterations of proteostasis, including a broad spectrum of age-related brain disorders, known as neurodegenerative diseases, characterized by the accumulation of aberrantly folded proteins in the central nervous system[5,6]. These range from highly frequent disorders, such as Alzheimer's and Parkinson's diseases, to rather rare conditions such as amyotrophic lateral sclerosis and prion diseases. The latter, also known as transmissible spongiform encephalopathies, are caused by the conformational conversion of a cell surface glycoprotein, named the cellular prion protein (PrP), into an aggregated, pathogenic form (called PrP scrapie, or PrP$^{Sc}$) capable of propagating like an infectious agent (prion) by templating the structural conversion of its physiological counterpart[7,8]. Growing evidence indicate that prions exemplify a mechanism of protein-based propagation of information exerted by many other amyloids in pathological and physiological contexts[9,10]. Unfortunately, the absence of detailed information regarding misfolding processes, as well as ineffective clearance by the proteostasis machinery, have so far hampered the development of therapies for the vast majority of neurodegenerative disorders. Fundamental insights into the regulation of protein expression, folding, and misfolding could theoretically be provided by the full reconstruction of the conformational transitions underlying folding pathways. However, experimental approaches aimed at addressing the dynamics of protein folding are seriously affected by the trade-off between temporal and spatial resolutions of available biophysical techniques[11]. Computer-based technologies, such as Molecular Dynamics (MD) simulations, could, in principle, overcome these limitations and help to predict the evolution in time of protein conformations. In practice, however, classical MD simulations are not applicable to study many fundamental molecular processes, such as the cascade of events underlying the folding of a typical, biologically relevant polypeptide bigger than 80 amino acids and with folding times longer than a few milliseconds[12]. More advanced methods, such as metadynamics or replica exchange, have the advantage that, under certain conditions, can lead to an exact sampling of the protein conformations. However, these techniques are too computationally expensive to be applicable to study the folding of biologically relevant proteins. To solve this issue, we have developed algorithms that reduce computational costs by exploiting available experimental information of native structures[13]. The reliability of these methods to predict protein folding mechanisms has been extensively assessed against plain MD simulations and experiments[14]. Here, we designed a drug discovery paradigm (named PPI-FIT) based on the rationale of negatively regulating the level of a given protein by targeting folding intermediates. The PPI-FIT approach was tested on PrP, as compelling genetic and experimental evidence indicates that lowering the expression of this protein could produce therapeutic benefits in prion diseases without causing major side effects[15,16].

## Results

**Identification of a PrP folding intermediate**. The rationale underlying PPI-FIT is based on designing compounds against the most kinetically- and thermodynamically relevant folding intermediate of a given protein, in order to stabilize such an intermediate and inhibit its transition to the native form. In a cellular environment, a stabilized folding intermediate could be recognized by the protein folding quality control machinery as an improperly folded polypeptide and sent to degradation[17]. Therefore, the PPI-FIT approach aims at identifying compounds capable of post-translationally decreasing the level of a target protein. We sought to test this drug discovery paradigm on PrP, based on four aspects: (i) several previous reports indicate that at least one intermediate is generated along the folding pathway of PrP[18–20]; (ii) the genetic suppression of PrP in animal models has been shown to produce little side effects only later in life[21]. This conclusion is also supported by loss-of-function PRNP alleles observed in healthy human subjects[16]; (iii) strong experimental evidence demonstrates that lowering the expression of PrP would confer therapeutic benefits in prion diseases, and possibly other neurodegenerative disorders linked to the toxicity-transducing activity of the protein[15,22]; (iv) however, multiple previous failures to target the native conformation of PrP pharmacologically suggest that this protein could be an undruggable factor[23,24]. In light of these considerations, we decided to reconstruct the entire sequence of events underlying the folding pathway of this protein at an atomistic level of resolution. The protein consists of a flexible, N-terminal segment (residues 24–120, human sequence), and a structured, C-terminal domain (residues 121–231) comprising three α-helices and two short β-strands flanking helix-1. The N-terminal segment is intrinsically disordered, a feature precluding its use for in silico approaches[25]. Thus, as a reference structure, we used the globular domain of human PrP (residues 125–228, PDB 1QLX; Fig. S1). We generated nine unfolded PrP conformations by thermal unfolding and used rMD to produce 20 folding trajectories for each conformation (making a total of 180 trajectories plotted for their bidimensional probability distribution in Fig. 1a). A variational approach, called Bias Functional (BF), was then employed to define the most statistically significant folding pathway for each set of trajectories, leading to nine least biased trajectories (an example is shown in the Movie S1). Conformations residing within the observed energy wells in the bidimensional distribution were sampled from each least biased trajectory (Fig. 1b). The analyses revealed the existence of an on-pathway PrP-folding intermediate structurally close to the native state, and explored by all least biased pathways. Subsequent clustering analysis of the ensemble of conformers populating the intermediate state enabled the definition of an all-atom structure for the PrP-folding intermediate (Fig. S2). As compared with the PrP native conformation, this structure is characterized by a displaced helix-1 missing its docking contacts with helix-3, leaving several hydrophobic residues, natively buried inside the protein core, exposed to the solvent (e.g., Y157, M206, V209; Fig. 1c).

**In silico identification of high-affinity ligands for the PrP folding intermediate**. We employed in silico modeling and virtual screening techniques to identify small ligands for the PrP folding intermediate. First, we searched for druggable binding sites unique in the structure of the latter and not present in the native PrP conformation. This step was carried out by means of SiteMap (Schrödinger Release 2017-4) and DoGSiteScorer

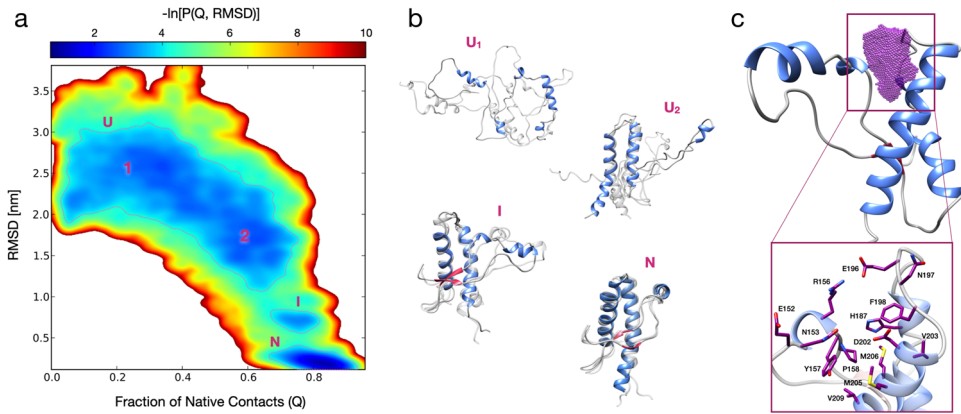

**Fig. 1 All-atom reconstruction of the PrP folding pathway. a** Lower-bound approximation of the free energy landscape of the PrP folding obtained from 180 rMD trajectories, plotted as a negative logarithm of the probability distribution expressed as a function of the collective variables Q (fraction of native contacts) and RMSD. The dashed lines delimit the metastable regions of interest (G ≤ 3.7 $k_B$T). **b** Representative PrP structures of unfolded (U1 and U2), folding intermediate (I) and native conformations (N). Each conformation represents the cluster center of one of the corresponding populated regions (structures depicted in transparency are sampled from the same region, showing conformations variability): the first two (corresponding to the unfolded states U1 and U2) are characterized by RMSD >1.8 nm and a Q < 0.5 (U1), and 1.2 < RMSD < 1.8 nm and a 0.5 < Q < 0.75 (U2); a third one (corresponding to the intermediate folding state I) with 0.55 < RMSD < 0.90 nm and a 0.65 < Q < 0.85; a fourth one (corresponding to the native state N) with RMSD < 0.40 nm and Q > 0.80. **c** Ribbon diagram of the PrP intermediate highlighting the ligand-binding pocket (purple dots) as identified by SiteMap and DogSiteScorer tools. The purple volume maps the unique druggable pocket identified in the PrP-folding intermediate I. The box shows specific residues defining the site.

software. The results highlighted the presence of a possible ligand-binding region, placed between the beginning of helix-1 and the loop that connects the helix-2 with helix-3 (Table S1). MD simulations refined the structure of a solvent-accessible, druggable pocket (Table S1) defined by 14 non-continuous residues (152, 153, 156, 157, 158, 187, 196, 197, 198, 202, 203, 205, 206, 209) (Fig. 1c). This site was the target of a virtual screening campaign performed by using the BioSolveIT software and employing the Asinex Gold & Platinum collection of small molecules (∼3.2 × 10⁵ commercially available compounds). The obtained poses were filtered according to the predicted estimated affinity, physicochemical, and ADMET (absorption, distribution, metabolism, excretion, and toxicity) properties as well as 3D visualization/assessment of the ligand-binding interactions. In addition, molecules potentially acting as pan-assay interference compounds were filtered, and cheminformatics methods for similarity and clustering analysis were applied to support a diversity-based selection. Finally, 30 compounds were selected as promising virtual hits and purchased for biological validation (Table S2 and Fig. S3).

**In vitro validation of in silico hits.** PrP biogenesis follows a trafficking pathway typical of glycosylphosphatidylinositol (GPI)-anchored polypeptides[26]. The protein is synthesized directly in the lumen of the endoplasmic reticulum (ER), where it folds and receives post-translational processing of the primary structure (removal of the signal peptide and anchoring of the GPI moiety) as well as the addition of two N-linked glycans (at Asn-181 and Asn-197)[27]. Based on the role of lysosomes in the quality control of PrP loads, a compound binding to a PrP folding intermediate may produce a long-living, immature conformer that could be recognized by the ER quality control machinery, leading to its degradation by the ER-associated lysosome-dependent autophagy[28]. Following this principle, the 30 putative ligands were directly tested in cells for their ability to lower the amount of PrP at a post-translational level. Each compound was administered for 48 h at different concentrations (1–30 μM) to HEK293 cells stably expressing mouse PrP. The expression and post-translational alterations of PrP were detected by western blotting. Compounds

showing an ability to lower the amount or alter the post-translational processing of PrP (≥30%) at any of the concentrations were tested against a control protein, the Neuronal Growth Regulator 1 (NEGR-1), a member of the immunoglobulin LON family. This GPI-anchored molecule follows the same biosynthesis pathway as PrP, thus representing an ideal control to evaluate compound specificity[29]. We validated four compounds (coded SM930, SM940, SM950, and SM875), all of them capable of decreasing the levels of PrP in HEK293 cells without lowering the control protein NEGR-1 (Fig. 2a–d). Among these, compound SM875 was selected for further analyses, based on its potency (docking poses of the compound bound to the PrP folding intermediate are shown in Fig. 2e).

**Synthesis and biological characterization of SM875.** In order to obtain large amounts of the compound for additional assays, we designed and performed a synthesis scheme for SM875 (Fig. S4). The correct structure of the molecule was verified by nuclear magnetic resonance (NMR), mass spectrometry (MS), and infrared (IR) spectroscopy (Fig. S5A–D and Table S3). The dose-dependent, PrP-lowering activity of the newly synthesized SM875 molecule was validated using a larger concentration range (0.01–50 μM) in stably transfected HEK293 cells, allowing the calculation of the inhibitory concentration at 50% in the low micromolar range (IC₅₀ = 7.87 ± 1.17 μM; Fig. 2a). Next, to rule out possible artifacts due to the exogenous transfection of PrP, we tested the effect of SM875 in a human breast cancer cell line (ZR-75) endogenously expressing human PrP. These cells were identified among the NCI collection of human tumor cell lines as high PrP expressors (Fig. S6). Compound SM875 decreased PrP levels in a dose-dependent fashion also in these cells, in the exact same concentration range (1–30 μM) observed in transfected HEK293 cells (Fig. 3a). Importantly, SM875 did not alter the level of Thy-1, another GPI-anchored protein endogenously expressed in ZR-75 and used as a control (Fig. 3a)[30]. Consistent with these data, SM875 reduced the level of PrP in two additional untransfected cell lines, L929 mouse fibroblasts, and mouse N2a neuroblastoma cells (Fig. 3b, c). Interestingly, in N2a cells SM875 did not show a dose–response effect, as PrP decrease was observed only at lower

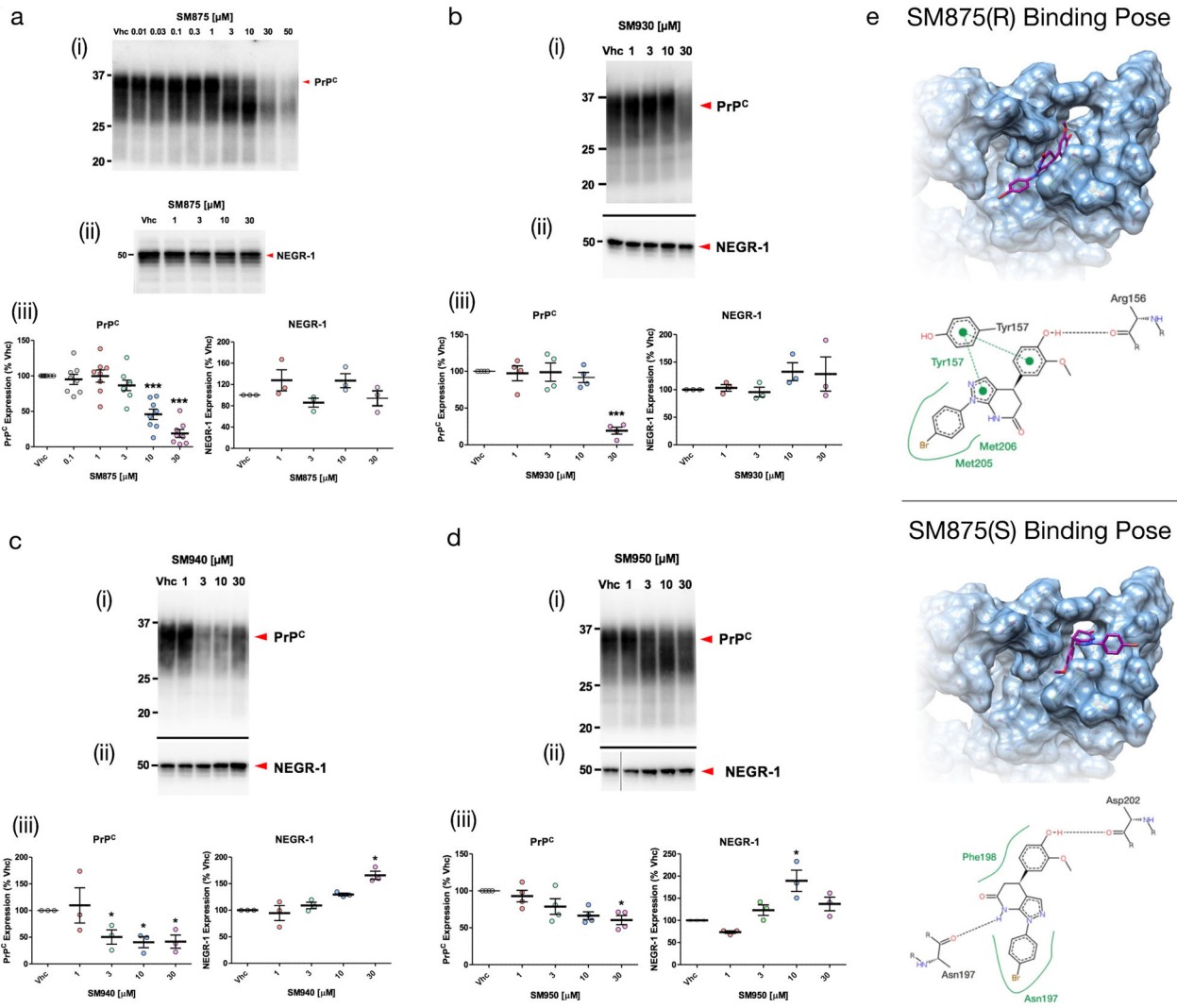

**Fig. 2 In vitro validation of selected hits.** HEK293 cells expressing mouse PrP or NEGR-1 were exposed to different concentrations of SM875 **a**, SM930 **b**, SM940 **c**, or SM950 **d** or vehicle (0.1% DMSO, volume equivalent) for 48 h, lysed in detergent buffer (Tris 10 mM, pH 7.4, 0.5% NP-40, 0.5% TX-100, 150 mM NaCl plus EDTA-free Protease Inhibitors), diluted in Laemmli sample buffer and analyzed by western blotting using anti-PrP (D18) or anti-NEGR-1 (R&D, USA) antibodies. Red arrowheads indicate the expected sizes of mature, fully glycosylated forms of PrP and NEGR-1. The compounds induce a dose-dependent suppression of PrP (i) but not control protein NEGR-1 (ii). The graphs (iii) show the densitometric quantification of the levels of full-length PrP or NEGR-1 from different biologically independent replicates. Each signal was normalized on the corresponding total protein lane (detected by UV of stain-free gels) and expressed as the percentage of the level in vehicle (Vhc)-treated controls (*$p < 0.05$, **$p < 0.01$, ***$p < 0.005$, by one-way ANOVA test). **e** The picture illustrates the predicted ligand-binding pose of the R (upper panel) and S (lower panel) SM875 enantiomers into the PrP intermediate druggable pocket.

concentrations (1–10 μM), whereas the protein showed normal levels at the highest concentration (30 μM). In order to address this discrepancy, we assessed the levels of PrP mRNA by quantitative real-time PCR (RT-PCR). We found that treatment with SM875 did not decrease PrP mRNA in the different cell lines even at the highest concentrations, confirming the original rationale that a compound targeting a folding intermediate of PrP should lower its expression at a post-translational level (Fig. 3d). Importantly, only in N2a cells, we observed a robust (>500% at 30 μM), dose-dependent increase of PrP mRNA upon treatment with SM875, which directly explains the discrepancy observed at the protein level. This observation also suggests that the compound triggers an over-production of PrP mRNA in these cells, likely representing a compensatory response aimed at restoring PrP levels. The compound showed variable intrinsic toxicity in the different cell lines, with more prominent cytotoxicity in PrP-transfected HEK293 cells, and progressively lower toxicity in

untransfected HEK293, N2a, ZR-75, and L929 cells, respectively (Figs. S7, S8). To further corroborate the observed effects of SM875 on PrP load, we checked whether the compound also decreases the amount of the protein at the cell surface. HEK293 cells stably transfected with an EGFP-tagged PrP construct were exposed to SM875, and the localization of PrP was monitored by detecting the intrinsic green fluorescence of EGFP. In control conditions, EGFP-PrP localizes almost entirely in the Golgi apparatus and at the plasma membrane, with the latter giving rise to a typical "honeycomb-like" staining of the cell surface (Fig. 3e) [31]. Compounds altering PrP trafficking, as the phenothiazine derivative chlorpromazine, have previously been shown to alter such localization pattern[32]. Incubation with SM875 for 24 h induced a drastic reduction of cell surface EGFP-PrP at concentrations as low as 1 μM. Collectively, these data confirm that SM875 selectively reduces the amount of PrP at the post-translational level and in a cell-independent fashion.

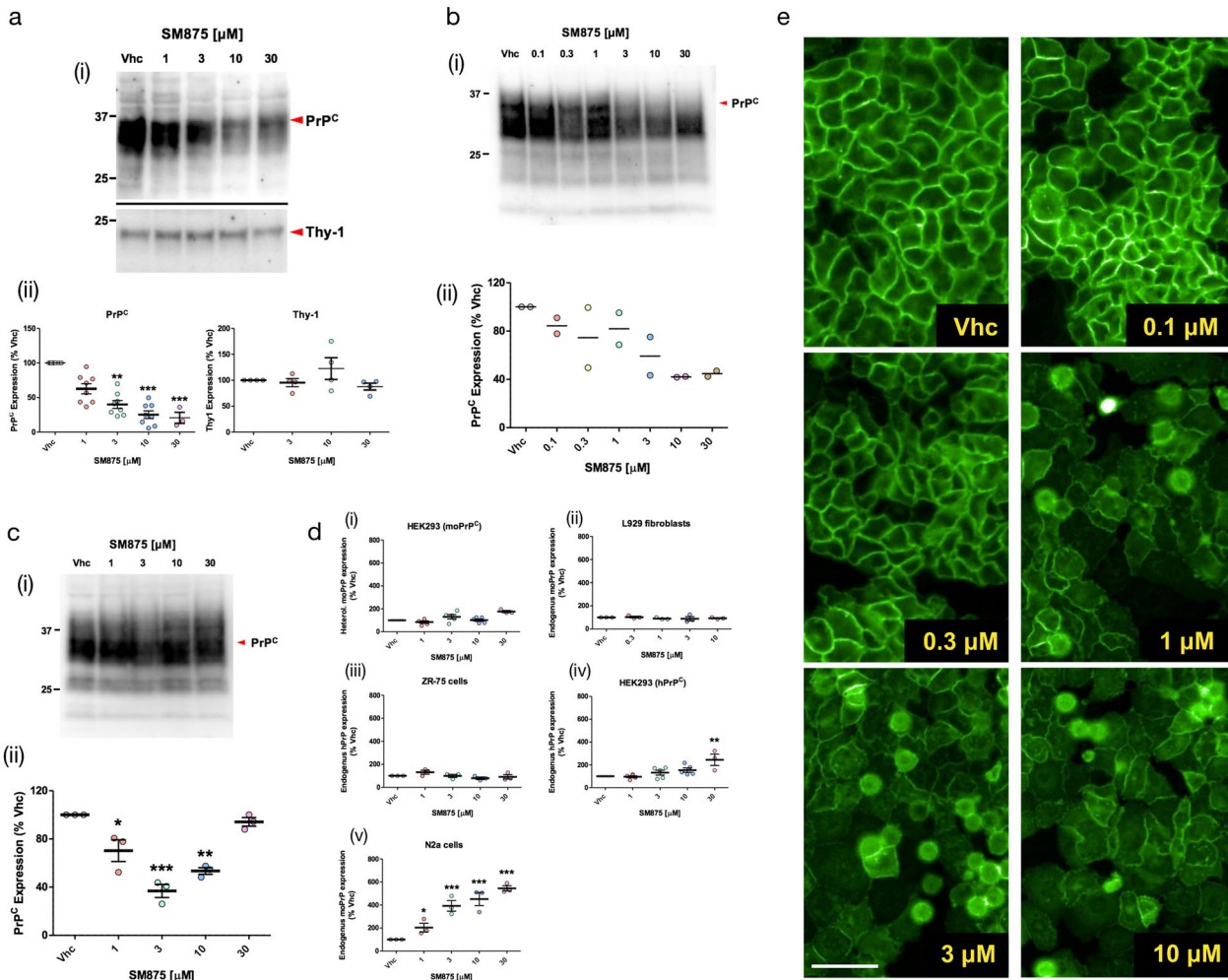

**Fig. 3 SM875 lowers the amount of PrP at a post-translational level in different cell lines.** Cells were exposed to different concentrations of SM875 or vehicle (0.1% DMSO) for 48 h, lysed, and analyzed by western blotting. **a** In ZR-75 cells, SM875 suppresses PrP, but not Thy-1, in a concentration-dependent fashion. **b** Similar effects were observed in cultured L929 fibroblasts. **c** In N2a cells, SM875 shows a dose-dependent lowering effect of PrP at 1–10 μM. However, in contrast to the other cell lines, the compound showed no effect at 30 μM. All signals were detected by using a specific anti-PrP (D18) or anti-Thy-1 primary antibodies. Red arrowheads indicate the expected sizes of mature, fully glycosylated forms of PrP or Thy-1. Western blotting analysis (i) and graphs reporting the densitometric quantification of signals (ii) are shown. Each signal was normalized on the corresponding total protein lane (detected by UV of stain-free gels) and expressed as the percentage of vehicle (Vhc)-treated controls (*$p < 0.05$, **$p < 0.01$, ***$p < 0.005$, by one-way ANOVA test). **d** Graphs show the levels of PrP mRNA upon treatment with SM875, as evaluated by RT-PCR. Specific forward and reverse primers were used to amplify endogenous or exogenous, mouse or human PrP transcripts (see Materials and Methods). Relative quantification was normalized to mouse or human HPRT (hypoxanthine–guanine phosphoribosyltransferase). Statistical analyses refer to the comparison with vehicle controls (**$p < 0.01$, ***$p < 0.005$, by one-way ANOVA test). Dots represent biologically independent replicates. **e** HEK293 stably expressing a PrP form tagged with a monomerized EGFP molecule at its N-terminus (EGFP-PrP) were incubated with vehicle (0.1% DMSO) control (i) or SM875 at different concentrations (ii-vi) for 24 h. Fluorescence of the EGFP protein was then visualized with an Olympus BX51WI microscope equipped with reflected fluorescence. Scale bar 50 μm.

**SM875 induces the degradation of PrP by the lysosomes.** Proteins trafficking through the ER that fail to fold properly are usually degraded to prevent the accumulation of toxic aggregates. Most of these aberrantly folded polypeptides undergo retro-translocation into the cytosol, poly-ubiquitination, and degradation by the 26S proteasome, a process known as ER-associated degradation (ERAD)[33]. Although ERAD is the main degradation pathway for most misfolded ER proteins, GPI-anchored proteins like PrP are primarily cleared by the so-called ER-to-lysosome-associated degradation pathway[34,35]. For example, the expression of disease-associated, aggregation-prone mutants of PrP in different cells increases several markers of the autophagy–lysosomal pathway, such as the autophagosome-specific marker microtubule-associated protein 1 A/1B-light chain 3-II (LC3-II)[28]. Here, we sought to test whether SM875 lowers PrP levels by inducing its lysosomal

degradation. By comparing PrP-transfected and untransfected HEK293 cells, we observed that SM875 (10–30 μM) increases LC3-II levels in a PrP-dependent manner (Fig. 4a). Next, we directly tested the role of lysosomal clearance for the observed PrP-lowering effect of SM875. As previous data showed that inhibiting degradation pathways in transfected cells could possibly generate over-transcriptional artifacts of exogenous genes[35,36], for these experiments, we relied on ZR-75 cells, expressing PrP endogenously. We found that autophagy inhibitor Bafilomycin A1 largely rescues SM875-induced PrP decrease in these cells (Fig. 4b). Of note, the natural disaccharide trehalose, reported to increase de novo formation of autophagosomes, did not alter the PrP-lowering effects of SM875 (Fig. S9)[37]. Collectively, these data indicate that SM875 promotes the degradation of PrP by the autophagy–lysosomal pathway.

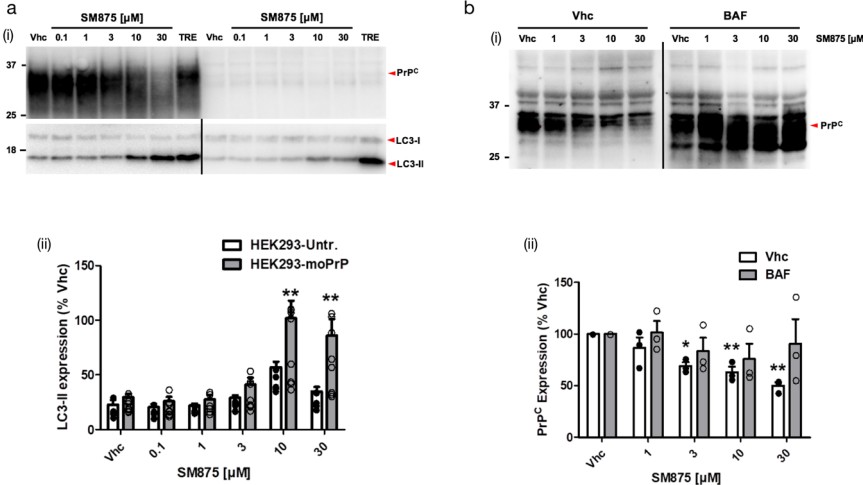

**Fig. 4 SM875 decreases PrP level via the autophagy-mediated lysosomal degradation. a** (i) PrP-transfected or untransfected HEK293 cells were treated with different concentrations of SM875 or vehicle (0.1% DMSO, volume equivalent) and the levels of PrP, LC3-I, and LC3-II were evaluated by western blotting (indicated by the red arrowheads). As a positive control, cells were treated with 100 µM trehalose (TRE), a disaccharide previously reported to induce autophagy. (ii) Graphs show the densitometric quantification of LC3-II from independent replicates ($n = 8$ for PrP-expressing HEK293 cells; $n = 5$ for untransfected HEK293 cells). Each signal was normalized on the corresponding total protein lane (detected by UV) and expressed as the percentage of vehicle (Vhc)-treated controls (**$p < 0.01$, by one-way ANOVA test). **b** (i) ZR-75 cells were treated with different concentrations of SM875 in the presence or absence of autophagy–lysosomal inhibitor bafilomycin A1 (BAF, 10 µM) and PrP levels were evaluated by western blotting. (ii) Graphs show the densitometric quantification of full-length PrP from independent replicates ($n = 3$). Each signal was normalized on the corresponding total protein lane (detected by UV) and expressed as the percentage of vehicle (Vhc)-treated controls (**$p < 0.01$, by one-way ANOVA test).

**SM875 acts exclusively on nascent PrP.** Based on the initial hypothesis that SM875 acts on an intermediate appearing along the folding pathway of PrP, we decided to rule out the possibility that the molecule instead targets native PrP. First, we used dynamic mass redistribution (DMR), a technique previously employed to characterize small molecule ligands of PrP. In contrast to $Fe^{3+}$-TMPyP, an iron tetrapyrrole known to interact with PrP, SM875 (0.1–100 µM) showed no detectable binding to either human full-length (23–231) or mouse N-terminally truncated (111–230) recombinant PrP molecules (Fig. 5a). These data indicate that the compound has no detectable affinity for natively folded PrP. In order to further validate this conclusion in a cell system, we turned to previously described RK13 cells expressing mouse PrP in a doxycycline-inducible fashion (Fig. S10)[38]. In the first set of experiments, we turned on the expression of PrP with doxycycline, in the presence of SM875 (10 µM), brefeldin-1A (10 µM), an inhibitor of the ER-to-Golgi protein trafficking[39], or vehicle control (0.1% dimethyl sulfoxide (DMSO), volume equivalent). Cells were then lysed and analyzed by western blotting at different time points (2, 4, 8, or 24 h, Fig. 5b panel i; Fig. S11A). As expected, in contrast to the typical size of full-length, diglycosylated PrP (~35 kDa) obtained in the control samples, brefeldin-1A induced the accumulation of an immature, low molecular weight band (~20 kDa), previously described to correspond to a N-terminally cleaved PrP molecule formed by a lysosomal-dependent event[40]. Interestingly, SM875 induced the accumulation of a similar PrP band, compatible with its hypothesized activity of targeting a folding intermediate of the protein (Fig. 5b panel I; and Fig. S11A). Next, we sought to directly test the effect of the molecule on pre-synthesized, mature PrP. To pursue this objective, we induced the expression of PrP for 24 h with doxycycline. We then removed the inducer, waited 4 h, treated the cells with SM875 or vehicle control, and analyzed PrP content at different time points (5, 19, and 24 h). In this case, we found no difference between compound- and control-treated samples (Fig. 5b, panel ii; and Fig. S11B). The results were further validated by high-content imaging analysis of immunostained

PrP, which showed that in the same doxycycline-induced RK13 cells, SM875 causes the rapid (as early as 4 h) accumulation of PrP species in intracellular compartments, preventing their correct delivery to the cell surface (Fig. 5c). Collectively, these data show that SM875 acts exclusively on nascent, immature PrP molecules, whereas the compound exerts no effect on pre-synthesized, mature PrP.

**SM875 suppresses prion replication in mouse L929 fibroblasts.** One of the most solid concepts in prion diseases is that suppressing PrP should abrogate prion replication[41]. The most direct way to achieve this objective is to silence PrP expression, either through a selective decrease of its synthesis or increase of its clearance. Following the observation that SM875 lowers PrP level by inducing the degradation of a folding intermediate, we tested the ability of the compound to inhibit the replication of the Rocky Mountain Laboratories (RML) prion strain in persistently infected L929 mouse fibroblasts[42]. We found that SM875 inhibits prion replication in a dose-dependent fashion, decreasing prion loads similarly to anti-prion molecule $Fe^{3+}$-TMPyP, used as a control (Fig. 6)[43].

**SM875 induces the aggregation of partially denatured PrP.** Despite the different pieces of evidence collected in silico and in cell-based assays, the formal demonstration that SM875 directly targets a folding intermediate of PrP would require a structural characterization of the complex. Unfortunately, solving the structure of folding intermediates is limited by the transient nature of these protein conformers. In an attempt to overcome this problem, we designed an experimental paradigm aimed at inducing the appearance of non-native conformers of PrP by thermally induced partial denaturation (Fig. S12). The underlying hypothesis is that the PrP folding intermediate predicted by our in silico analyses could also be reached from the native state by overcoming an energy barrier so that SM875 could bind and stabilize it. Here, we explored this possibility by trying to

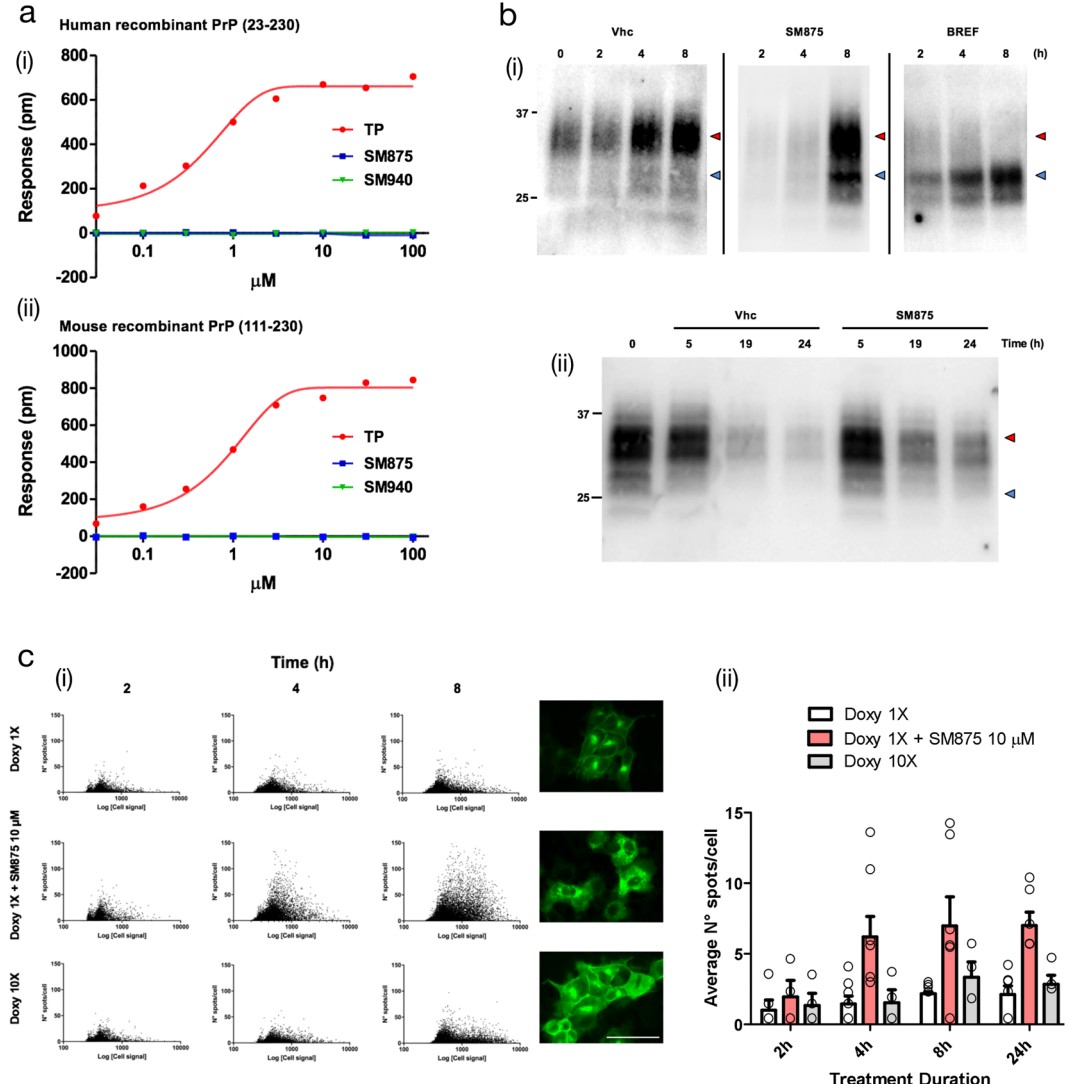

**Fig. 5 SM875 acts exclusively on non-native, newly synthesized PrP. a** DMR was employed to assess whether SM875 has an affinity for the native conformation of PrP. Different concentrations (0.03–100 μM) of SM875, SM940 or PrP ligand Fe3+-TMPyP (TP), used as a control, were added to label-free microplate well surfaces on which either (i) full-length, human recombinant PrP (23–230) or (ii) N-terminally deleted mouse recombinant PrP (111–230) had previously been immobilized. All signals were fitted (continuous lines), when possible, to a sigmoidal function using a 4PL non-linear regression model. In contrast to SM875 and SM940, TP shows a detectable affinity for both full-length, human, and N-terminally deleted, mouse PrP molecules (for full-length PrP, $K_d = 0.67 \pm 0.05$, $R^2 = 0.99$). **b** In order to dissect the effect of SM875 on nascent vs mature, native PrP molecules, we turned to RK13 cells expressing mouse PrP under control a doxycycline-inducible promoter. (i) PrP expression was induced over 8 h, in the presence of SM875 (10 μM), brefeldin-1A (BREF, 10 μM), or vehicle (Vhc) control, samples were collected at different time points (indicated) and PrP signals were visualized by western blotting. Signals were detected by probing membrane blots with anti-PrP antibody (D18). As expected, in control cells, the level of full-length PrP (red arrowheads) increases in a time-dependent fashion. Conversely, a lower molecular weight band (blue arrowheads) is detected in brefeldin-treated cells. (ii) Next, we designed an experiment to test the effect of SM875 exclusively on mature, natively folded PrP. PrP expression was induced for 24 h, in the absence of any additional treatment. Doxycycline was then removed, and after 4 h without inducer, the cells were exposed to SM875 or Vhc control, and subsequently lysed at different time points (indicated). In this experimental setting, cells are exposed to SM875 only when all PrP molecules are synthesized and likely in transit to, or already reached the plasma membrane. In these conditions, normal PrP patterns appear in both compound-treated and Vhc-treated cells. **c** A high-content approach was employed to the same experimental setting described above by analyzing the localization of PrP after immunostaining with an anti-PrP antibody (D18) coupled to an Alexa 488 secondary antibody. (i) The expression of PrP was induced for 2, 4, or 8 h by doxycycline 1× (0.01 mg/mL) or 10× (0.1 mg/mL) and the effect of SM875 (10 μM) incubated with doxycycline 1× was measured by Harmony software after the image acquisition performed by Operetta Imaging System. Green spots detected in cells were quantified and plotted against the total green fluorescence relative to each cell. Representative images were acquired at 8 h of incubation, scale bar 50 μm. (ii) Quantification of the average number of green spots per cell in wells incubated for 2, 4, 8, and 24 h with doxycycline 1× (white bars), doxycycline 10× (gray bars) and doxycycline 1× + SM875 (red bars). Quantification of at least three independent experiments.

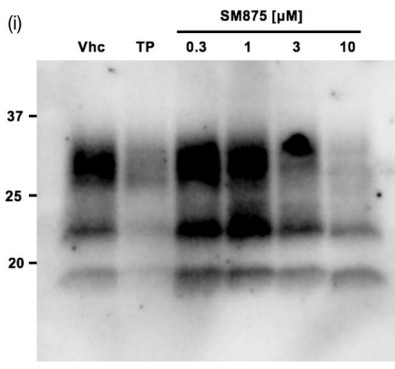
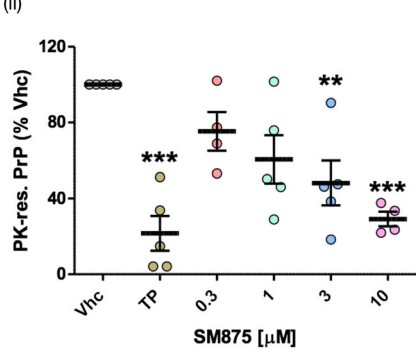

**Fig. 6 SM875 inhibits prion replication in L929 mouse fibroblasts.** L929 mouse fibroblasts were infected with the RML prion strain and then propagated for five sequential passages before exposure to SM875 (indicated concentrations), anti-prion compound $Fe^{3+}$-TMPyP (TP, 10 μM) or vehicle (0.1% DMSO) for 48 h. $PrP^{Sc}$ loads were then estimated by treating cell lysates with PK and analyzing PrP content by western blotting (i). Signals were detected by probing membrane blots with anti-PrP antibody (D18). Results show that SM875 inhibits prion replication in a dose-dependent fashion, with the maximal effect (obtained at 10 μM) comparable to that of TP; (ii) the graph shows the densitometric quantification of all PK-resistant PrP bands from independent replicates (n ≥ 4). Quantification was obtained by densitometric analysis of the western blots, normalizing each signal on the corresponding PK-untreated lane and expressed as the percentage of vehicle (Vhc)-treated controls (** $p < 0.01$, *** $p < 0.005$, by one-way ANOVA test).

crystallize the temperature-induced, semi-denatured PrP in complex with SM875. Recombinant mouse PrP (800 μM) was heated to 45 °C and slowly cooled to 20 °C using a thermal cycler, either in the presence or absence of SM875 (2 mM). The high concentrations of both protein and compounds were compatible with the crystallization process. Unexpectedly, we observed massive precipitation (~40%) of the protein as soon as SM875 was added to the solution, as assayed by UV absorbance. Conversely, no appreciable precipitation was detected in vehicle-treated controls. The remaining soluble PrP molecules gave rise to "thin needles"-like crystals, appearing after 2–4 days. Importantly, such crystals were obtained irrespectively of whether the protein was incubated with SM875 or vehicle. Crystals were weakly and anisotropically diffracting to 3.7 Å in the best directions, allowing determination of orthorhombic crystal system and unit cell dimensions, highly similar to those reported for apo human PrP in the protein databank (PDB 3HAK) (Table S4). These data suggest that SM875 targets a thermally induced PrP population, promoting its precipitation. Conversely, the more native-like PrP molecules do not bind to SM875 and remain in solution, allowing crystallization. The observed precipitation of partially denatured PrP induced by SM875 could reflect an intrinsic propensity of a PrP folding intermediate to expose hydrophobic residues normally buried in the native state. In order to better characterize this effect, we performed a detergent insolubility assay to detect insoluble aggregates of mouse recombinant PrP (111–230) in the presence or absence of SM875 upon temperature shift (25, 37, 45, 55 °C; the reported melting temperature for recombinant PrP is ~65 °C)[44]. Of note, the concentrations of the protein (0.5 μM) and compound (10 μM) were substantially lower in this assay as compared to those used for crystallization. We found that, differently from vehicle-treated samples, SM875 induced the aggregation of recombinant PrP in a temperature-dependent fashion (Fig. 7a). Similar results were obtained with compound SM940 but not SM935, found positive and negative for PrP downregulation in cells, respectively (Fig. 3c and Fig. S13). Interestingly, direct visualization of SM875-induced PrP aggregates by field emission scanning electron microscopy (FESEM) revealed the presence of amorphous structures appearing as a multitude of dots showing different sizes and shapes (~10 nm diameter) as well as much larger clumps (Fig. S14). Sonication did not affect the morphology of these aggregates, leading only to the breakage of the larger structures. Despite being at low resolution, these results suggest that binding to SM875 induces the formation

of aggregated PrP species that are off-pathway of the folding process. Importantly, plain MD simulations (21 μs of cumulative simulation time, Fig. S15) revealed the appearance of a conformation highly resembling the folding intermediate occasionally explored by the native PrP state when subjected to a temperature higher than RT (310 K; Fig. 7b). Collectively, these data add direct evidence for the interaction of SM875 with a non-native PrP conformer that could be explored by the natively folded PrP in a temperature-dependent fashion.

## Discussion

We have built on advanced computational techniques to test an entirely novel method, to the best of our knowledge, for selectively reducing the level of target proteins, which we called PPI-FIT. This technology is based on the concept of targeting folding intermediates of proteins rather than native conformations. In the PPI-FIT method, druggable pockets appearing in specific folding intermediates observed along the folding pathway of a given protein are used as targets for virtual screening campaigns aimed at identifying small ligands for such regions. The underlying rationale is that stabilizing a folding intermediate of a protein with small ligands could promote its degradation by the cellular quality control machinery, which could recognize such artificially stabilized intermediates as improperly folded species. In this manuscript, we applied this technology to PrP, identified a specific intermediate along the folding pathway of the protein, and then carried out a virtual screening campaign that led to the discovery of four different chemical scaffolds all capable of selectively lowering the level of PrP. Extensive characterization of one of these molecules, called SM875, provided strong experimental support for the notion that targeting folding intermediates could represent a pharmacological paradigm to modulate protein loads. From a broader perspective, our data reveal the existence of a previously unappreciated layer of regulation of protein expression occurring at the level of folding pathways.

The demonstration that the total level of a protein could be regulated by targeting a folding intermediate, the concept underlying the PPI-FIT method, would ultimately require a direct biophysical characterization of the interaction between the identified ligands and the predicted folding conformer. Unfortunately, available high-resolution techniques such as X-ray crystallography or NMR can only be applied to stable molecular species, while protein folding intermediates are transient and

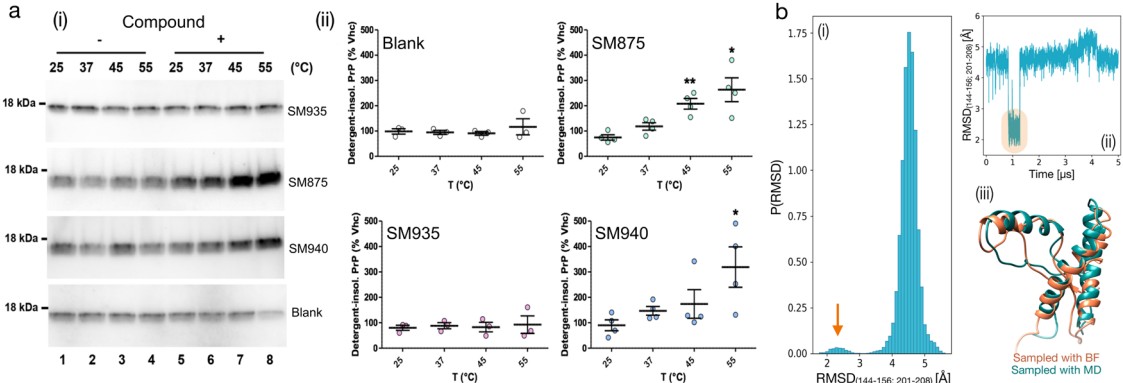

**Fig. 7 SM875 induces the aggregation of PrP in a temperature-dependent fashion. a** Recombinant PrP (111–231) was diluted in a detergent buffer (final concentration of 0.5 µM), placed at one of the indicated temperatures and incubated for 1 h with either vehicle (Vhc, lanes 1–4) or (lanes 5–8) assay buffer (blank), SM875, SM940, or SM935 (10 µM). After incubation, samples were subjected to ultracentrifugation, and the resulting detergent-insoluble pellets, corresponding to the aggregated fraction of PrP, were analyzed by western blotting (i). Signals were detected by probing membrane blots with anti-PrP antibody (D18). (ii) Graphs show the densitometric quantification of recombinant PrP bands from independent replicates. Each signal was normalized and expressed as the percentage of the corresponding Vhc-treated sample (*$p < 0.05$, **$p < 0.01$, by one-way ANOVA test). **b** Native PrP (residues 125–228, PDB 1QLX) was subjected to 21 µs of cumulative ($3 × 5$ µs + $2 × 3$ µs) MD simulations at 310 K. (i) The RMSD of residues lying within the contact region between helix-1 and helix-3 (indicated) was computed with respect to the structure of the previously identified PrP-folding intermediate. (i) The graph shows the probability distribution of the RMSD. The orange arrow indicates the population of PrP conformations resembling the folding intermediate. (ii) Individual trajectory exploring the PrP-folding intermediate from the native state (highlighted in orange). (iii) Superimposition of the PrP-folding intermediate identified by the BF method and the conformation reached from the native state, as observed by plain MD.

highly dynamic by definition[45]. Conversely, techniques designed to study single molecules, like optical tweezers, or to capture very rapid molecular transitions, such as spectroscopic detection coupled to stopped-flow methods, are intrinsically low resolution. In fact, the existence of a folding intermediate of PrP was previously detected by using similar approaches but never characterized at a structural level[18–20]. Despite the intrinsic approximations of all in silico techniques, our computational platform overcomes the spatiotemporal resolution limits, allowing us to define the atomistic structure of the PrP folding intermediate. Interestingly, the same conformation was occasionally observed by plain MD simulations of native PrP at 310 K, ruling out possible biases introduced by the BF method. We used this information to carry out a virtual screening campaign aimed at identifying small ligands for such PrP folding conformer. Then, in light of the aforementioned experimental limits, we designed an extensive in vitro and cell-based workflow to validate the PPI-FIT method. First, we identified four compounds, displaying different chemical scaffolds, all capable of selectively lowering the amount of PrP. Next, we found that SM875, the most potent among the positive hits, reduces PrP loads in different cell lines, either expressing the protein endogenously or exogenously, without decreasing PrP mRNA levels. Our data also show that SM875 promotes the degradation of nascent PrP molecules by the autophagy–lysosomal pathway, whereas the molecule neither binds nor exerts any effect on mature, cell surface PrP. Finally, in an attempt to co-crystallize the PrP-folding intermediate bound to SM875, we found that the small molecule induces the precipitation of PrP species appearing upon mild thermal unfolding. Collectively, even though we do not formally demonstrate the interaction of SM875 with the predicted binding pocket, our data provide strong experimental support for the notion that the compound acts as by targeting a folding intermediate of PrP (Fig. 8).

Prion diseases represented a convenient experimental ground for testing the PPI-FIT approach, as compelling evidence indicates that prion replication and toxicity can be inhibited by targeting a single protein (PrP)[15]. Theoretical options to achieve this goal include the ablation of the PrP gene, for example by using

the CRISPR technology[46], decreasing its synthesis with specific antisense oligonucleotides (ASOs)[47], or acting directly on cell surface PrP[48], for example, with ligands stabilizing the native fold and inhibiting its conversion into PrPSc [24,44]. Despite encouraging results obtained with ASOs recently, so far, none of these approaches has been translated into effective therapies for prion diseases. Moreover, multiple unsuccessful attempts to identify small molecules specific for PrP suggest that this protein could be an undruggable target[23,24]. Conceptually, a compound like SM875 capable of selectively promoting the degradation of PrP by targeting a folding intermediate could be the ideal candidate to tackle prion diseases pharmacologically, and possibly synergize with other PrP-silencing strategies. Our data show that SM875 alters the correct maturation of the PrP polypeptide, similarly to ER-to-Golgi trafficking inhibitor brefeldin 1 A. Moreover, in prion-infected cells, the PrP-lowering activity of SM875 produces a strong inhibition of prion replication, comparable to that of the potent anti-prion compound $Fe^{3+}$- TMPyP. However, in contrast to brefeldin 1 A and $Fe^{3+}$-TMPyP, which are known to be rather non-specific[49], the effect of SM875 is selective for PrP, as demonstrated by the lack of alterations of other two GPI-anchored proteins (NEGR-1 and Thy-1). These results indicate that the further development of PPI-FIT-derived molecules into compounds administrable in vivo and their subsequent validation in animal models of prion diseases could represent fundamental steps to move these classes of anti-prion compounds to the clinical phase.

A major virtue of the PPI-FIT approach is that it capitalizes on the cellular quality control machinery to promote the degradation of the target polypeptide. Such an effect is predicted to be obtained by targeting the folding process of a protein and it would not be achieved by putative binding events occurring with the same intermediate generated by the intrinsic dynamics of the native form. We envision at least two important limitations for the applicability of the PPI-FIT method to other proteins. First, the need for a high-resolution structure of the target, which is required for the computational reconstruction of the folding process. Moreover, the method is not applicable to proteins that fold by a chaperone-assisted mechanism, as this would

**Fig. 8 Model for the PPI-FIT-based suppression of PrP.** The model schematically illustrates the rationale underlying the PPI-FIT method applied to PrP. (i and iii) The schematics highlight PrP folding in the presence or absence of a small molecule targeting the PrP intermediate. (ii) PrP follows a typical expression pathway for GPI-anchored proteins. The polypeptide is directly synthesized into the lumen of the ER, once properly folded and post-translationally modified it traffics to the Golgi apparatus, where sugar moieties are matured, and then delivered to the cell surface. From the plasma membrane, PrP molecules could enter into the endosomal recycling pathways, and eventually recycled into the lysosomes. (iv) Aberrantly folded conformers of PrP, as the case for some previously reported mutant, could be re-routed directly from ER to the lysosomes. In the PPI-FIT method, a small molecule interferes with PrP expression by binding to a folding intermediate, which gets delivered to the lysosomes, leading to an overall decrease of the protein at the cell surface.

undermine the reliability of the folding process prediction as well as decrease the chance of identifying solvent-exposed pockets. Even considering these limitations, the PPI-FIT method may be useful in fields well beyond protein misfolding, as it could theoretically be applied to any protein for which the structure has been solved and not requiring chaperones to fold. Another possible drawback of PPI-FIT is represented by the location of the targets, which generally lie within intracellular compartments such as the ER. Thus, any molecule designed to act by PPI-FIT should possess the ability to cross lipid bilayers. From this standpoint, PPI-FIT would be no different than classical pharmacological approaches directed against intracellular targets or from other compounds acting as degraders. In fact, a similar rationale of PPI-FIT is exploited by emerging pharmacological strategies like the proteolysis targeting chimeras (PROTACs)[50]. This approach builds on the principle of designing bi-functional degraders composed by two covalently linked chemical moieties, one interacting with the target protein and the second engaging the E3 ubiquitin ligase, which then targets the polypeptide to degradation by the proteasome. Notably, two PROTAC compounds against the estrogen receptor and the androgen receptor have recently reached the clinical phase (trial numbers NCT03888612 and NCT04072952). However, compounds like PROTACs target native structures of proteins therefore suffering from the same limits as classical pharmacological approaches

when applied to undruggable targets like PrP. The PPI-FIT method theoretically expands the range of druggable conformations of a polypeptide by targeting alternative structures explored by the protein along its folding pathway. In fact, the PrP-folding intermediate provided a solvent-exposed, druggable pocket hidden in the native conformation. This feature may result in an additional advantage: a common problem in pharmacology is posed by proteins with high structural similarity of their native conformation, a factor that greatly decreases the chances of developing selective drugs. However, proteins with highly similar three-dimensional native folds do not necessarily share folding pathways, and even proteins with identical folding pathways may differ for the lifetimes of folding intermediates, which would then display different affinities for the same compound. Thus, the PPI-FIT approach could intercept structurally and/or kinetically different folding intermediates and ultimately allow the identification of selective ligands even in the case of highly similar proteins. Finally, it will be interesting to evaluate in the future whether existing drugs that lower the expression of specific proteins by unknown mechanisms could be acting in a PPI-FIT-like manner by targeting folding intermediates.

The observation that the expression of PrP could be modulated by altering the metabolism of non-native conformers of the protein appearing along the folding pathway suggests that similar mechanisms of regulation could be exploited in biology to

manipulate the levels of several other proteins. Interestingly, such a mechanism could be directly interconnected with the protein quality control machinery, potentially conferring an additional degree of complexity for the spatial and temporal regulation of proteostasis. Regulating the homeostasis of a folding intermediate of a protein, rather than just acting on transcription, translation, or clearance of its mature form, could be energetically favorable and allow the cell to respond to stimuli more rapidly. Similar mechanisms may play important roles in several other disease contexts beyond the one addressed here. For example, altering the folding pathways of specific homeostatic regulators or immunoregulatory factors could be a way for cancer cells to overcome cellular checkpoints, or for pathogens to evade host defenses. On an even broader perspective, we believe the existence of functional folding intermediates of proteins may imply that the evolution of polypeptide sequences does not act exclusively on native conformations, but also on alternative protein conformers transiently appearing along the folding pathways.

## Methods

### BF method for folding pathways simulations

*General features and software.* The BF method is a three-steps procedure that enables the simulation of protein folding pathways at atomistic level of resolution and consists of: (i) generation of denatured condition by thermal unfolding, (ii) productions of folding trajectories starting from the unfolded states and (iii) scoring of the folding trajectories based on a variational principle. The software to perform these simulations relies on the MD engine of Gromacs 4.6.5 patched with the plugin for collective variable analysis Plumed 2.0.2.

*Generation of denatured conditions.* Unfolded conformations are generated by thermal unfolding starting from the native structure. This is achieved by performing independent 3 ns trajectories of MD at high temperature (800 K) in the canonical (NVT) ensemble. For each trajectory, a single denatured conformation is extracted.

*Generation of folding pathways.* For each denatured conformation, a set of folding trajectories are generated by employing the ratchet and pawl molecular dynamics (rMD) algorithm. In this scheme, the folding progress is described as a function of a reaction coordinate, defined as z(X):

$$z(X) \equiv \sum_{|i-j|<35}^{N} \left[ C_{ij}(X) - C_{ij}(X^{\text{native}}) \right]^2 \quad (1)$$

Where $C_{ij}(X)$ is the contact map of the instantaneous system configuration and $C_{ij}(X^{\text{native}})$ Is the contact map in the reference native state. The reference native state is obtained by energy minimizing the experimental structure retrieved from the protein databank. The $C_{ij}(X)$ entries of z(X) interpolate smoothly between 0 and 1 according to the following function:

$$C_{ij}(X) = \begin{cases} \frac{1-(r_{ij}/r_0)^6}{1-(r_{ij}/r_0)^{10}} & if\ r_{ij} \leq r_c \\ 6/10 & if\ r_{ij} = r_0 \\ 0 & if\ r_{ij} > r_c \end{cases} \quad (2)$$

Where $r_{ij}$ is the Euclidean distance between the ith and the jth atom, $r_0$ is a typical distance defying residue contacts (set to 7.5 Å) and $r_c$ is a cutoff distance (set to 12.3 Å) beyond which the contact is set to 0. In rMD, the protein evolves according to plain MD as long as the reaction spontaneously proceeds towards the native state (i.e., lowering the value of the z(x) coordinate). On the other hand, when the chain tries to backtrack along z(X), an external biasing force is introduced that redirects the dynamics towards the native state. The biasing force acting on a given atom, $F_i^{rMD}$, is defined as:

$$\mathbf{F}_i^{rMD} = \begin{cases} -k_r \nabla_i z(X) \cdot [z(X) - z_m(t)] & if\ z(X) > z_m(t) \\ 0 & if\ z(X) \leq z_m(t) \end{cases} \quad (3)$$

Where $z_m(t)$ indicates the smallest value of the reaction coordinate z(X) up to time t and $k_r$ is a coupling constant.

*Selection of least biased trajectory.* For each set of trajectories starting from the same initial condition, the folding pathway with the highest probability to realize in the absence of external biasing force is selected. This scheme is applied by first defining a folding threshold: a trajectory is considered to have reached the folded state if its root mean squared deviation of atomic positions (RMSD) compared with the native target structure is ≤4 Å. Then, the trajectories successfully reaching the native state are scored by their computed BF T, defined as:

$$T = \sum_{i=1}^{N} \frac{1}{m_i \gamma_i} \int_0^t d\tau \left| \mathbf{F}_i^{rMD}(X, \tau) \right|^2 \quad (4)$$

Where t is the trajectory folding time, $m_i$ and $\gamma_i$ are the mass and the friction coefficient of the $i_{th}$ atom and $F_i^{rMD}$ is the force acting on it. The folding trajectory minimizing the BF for each set is referred to as Least Biased (LB) trajectory.

### Computational analysis of the cellular PrP

*Structure and topology of the C-terminal domain of the cellular PrP.* The native structure of the C-terminal domain of human PrP was retrieved from PDB 1QLX, the structure spans from residue 125 to 228 and contain the structured globular domain of PrP. Protein topology for folding simulations was generated in Gromacs 4.6.5 using Amber99SB-ILDN force field in TIP3P water. Protein topology for plain MD simulations was generated in Gromacs 2018 using Charmm36m force field in TIP3P water.

*Folding simulations of PrP.* The native structure of the C-terminal domain of human PrP (PDB 1QLX) was positioned in a dodecahedral box with 40 Å minimum distance from the walls. The box was filled with TIP3P water molecules and neutralized with three Na$^+$ ions. The system was energy minimized using the steepest descent algorithm. NVT equilibration was then performed for 500 ps at 800 K using the V-rescale thermostat with positional restraints on heavy atoms. Restraints were then removed and nine independent 3 ns of plain MD were performed in the NVT ensemble at 800 K, yielding nine denatured conformation. Each initial condition was repositioned in a dodecahedral box with 15 Å minimum distance from the walls, energy minimized using the steepest descent algorithm and then equilibrated first in the NVT ensemble (using the Nosé-Hoover thermostat at 350 K, τT = 1 ps) and then in the NPT ensemble (using the Nosé-Hoover thermostat at 350 K, τT = 1 ps, and the Parrinello–Rahman barostat at 1 bar, τP = 2 ps). For each initial condition, 20 trajectories were generated by employing the rMD algorithm in the NPT ensemble (350 K, 1 bar). Each trajectory consists in $1.5 \times 10^6$ rMD steps generated with a leap-frog integrator with time-step of 2 fs. Frames were saved every $5 \times 10^2$ steps. The ratchet constant $k_r$ was set to $5 \times 10^{-4}$ kJ/mol. Non-bonded interactions were treated as follow: Van-der-Waals and Coulomb cutoff was set to 16 Å, whereas Particle Mesh Ewald was employed for long-range electrostatics. For each set of trajectories, the BF scheme was applied with additional filtering on the secondary structure content for folding definition. In particular, trajectories reaching a final conformation with <85% of average secondary structure content compared to the NMR structure were not considered in the ranking.

*Analysis of the folding trajectories.* RMSD was computed using Gromacs while the fraction of native contacts (Q) was computed using VMD 1.9.2. A lower-bound approximation of the energy landscape G(Q, RMSD) was generated by plotting the negative logarithm of the 2D probability distribution of the collective variables Q and RMSD, obtained from the 180 rMD trajectories (115 × 115 bins). Protein conformations belonging to the LB trajectories and spanning over the energetic wells of interest (G ≤ 3.7 $k_B$T) were sampled. Conformations belonging to the intermediate state were clustered by using the k-mean approach in R-Studio[51] employing the following metrics for defining a distance between two structures:

$$D(X_A, X_B) = \sqrt{\sum_{|i-j|}^{N} \left[ C_{ij}(X_A) - C_{ij}(X_B) \right]^2} \quad (5)$$

Where $D(X_A, X_B)$ is the distance metrics between two protein conformations, $C_{ij}(X_A)$ and $C_{ij}(X_B)$ are the contact map entries of the conformations A and B respectively (defined in Eq. 2). The appropriate number of cluster (k = 3) was selected using the elbow method. The representative configuration of each cluster was selected by calculating the average contact map of the cluster conformations and then extracting the structure minimizing the distance $D(X_A, X_B)$ between itself and the average contact map. Data were represented using the Matplotlib library in python, the 2D [Q, RMSD] energy plot was smoothed with a Gaussian kernel. Images of the protein conformations were produced using UCSF Chimera.

*Plain MD simulations of native PrP.* The native structure of the C-terminal domain of human PrP (PDB 1QLX) was positioned in a dodecahedral box with 11 Å minimum distance from the walls. The box was filled with TIP3P water molecules, neutralized with counterions and brought to a final 150 mM NaCl concentration. The system was energy minimized using the steepest descent algorithm. NVT equilibration was then performed for 1 ns at 310 K using the V-rescale thermostat followed by 1 ns of NPT equilibration with the V-rescale thermostat and the Parrinello–Rahman Barostat at 310 K and 1 Bar (NVT and NPT equilibrations were carried out with positional restraints on heavy atoms). The equilibrations were repeated 5 times (each one starting from the energy minimized structure) giving rise to five initial conditions. For each initial condition, a single MD trajectory was launched (NPT, 310 K, 1 Bar), yielding a total of three trajectories (5 μs each) and two trajectories (3 μs each). The leap-frog algorithm (dt = 2 fs) was employed to perform these simulations. Non-bonded interactions were treated as

follow: Van-der-Waals and Coulomb cutoff was set to 12 Å, whereas Particle Mesh Ewald was employed for long-range electrostatics. A force-switch Van-der-Waals modifier with switch radius of 10 Å was used.

*Analysis of plain MD trajectories.* The Cα RMSD of residues residing in the contact region between helix-1 and helix-3 (144–156 and 201–208) was computed with respect to the structure of the PrP folding intermediate previously identified using the BF method. The RMSD computed for each nanosecond of trajectory was then used to generate a probability distribution of this quantity.

## Computer-aided drug discovery analyses

*Consensus approach for druggable ligand-binding site identification.* A consensus approach relying on SiteMap (Schrödinger Release 2017-4: SiteMap, Schrödinger)[52] and DoGSiteScorer[53] analysis has been applied to find and evaluate druggable binding pocket. Default parameters were used for both tools. Specific structural properties, namely volume, depth, enclosure/exposure and balance and different druggability scores were computed for each identified site. In particular, the exposure, enclosure and depth properties provide a different measure of how the site is prone to be a deep pocket. For the exposure property, the lower the score, the better the site, with the average value for tight-binding sites found to be 0.49. Conversely, higher scores are preferred for the enclosure descriptor, with the average enclosure score for a tight-binding site being 0.78. On the other hand, the balance property of SiteMap expresses the ratio between the relative hydrophobic and hydrophilic character of a pocket. This fraction has proven to be a highly discriminating property in distinguishing between druggable and undruggable pockets, with a mean value for tight-binding site equal to 1.6. Besides the global pocket descriptors, the applied tools provide automated methods for a quantitative estimation of druggability. SiteMap predicts a site score (SiteScore) and druggabilty score (DScore) through a linear combination of only three single descriptors: the size of the binding pocket, its enclosure, and a penalty for its hydrophilicity. The two scores differ in the coefficients, which are based on different training sets and strategies. A score value of ≥ 0.8 and ≥ 0.98 for SiteScore and DScore, respectively, has been reported to accurately distinguish between drug-binding and non-drug-binding sites. DoGSiteScorer also generates two scores, i.e., SimpleScore and a druggability score (DrugScore), which range from zero to one. The druggability cutoff for both scores is set to 0.5, indicating that targets with score above this value are considered as being druggable. Against this backdrop and considering the innovative character of the target herein reported (i.e., a folding intermediate structure), less stringent thresholds were thus selected in the search of druggable pockets to be explored in virtual screening: volume ≥ 300 Å³; depth ≥ 10 Å; balance ≥ 1.0; exposure ≤ 0.5; enclosure ≥ 0.70; SiteScore ≥ 0.8; DScore ≥ 0.90; DrugScore ≥ 0.5; SimpleScore ≥ 0.5 (Table S1).

*Identification of a druggable pocket in the PrP folding intermediate.* The previously described consensus approach allowed the identification of a possible ligand-binding region in the PrP folding intermediate, placed between the beginning of helix-1 and the loop that connects the helix-2 and helix-3 (Table S1). MD simulation were carried out to extract the most druggable conformation of this region, suitable for our virtual screening purpose. The folding intermediate was prepared with the Schrödinger's Protein Preparation Wizard. During the preparation, the hydrogen bonding networks were optimized through an exhaustive sampling of hydroxyl and thiol groups. The N- and C- terminal residues were capped with ACE and NMA groups, respectively. Then, hydrogen atoms and protein side chains were energy minimized using the OPLS3 force field. The obtained structure was (i) solvated by TIP3P water molecules in a cubic simulation box of 12.5 Å distant from the protein in every direction, (ii) neutralized by addition of three $Na^+$ ions, and (iii) equilibrated for 100 ps of MD simulation (NPT ensemble) at 300 K using the Langevin thermostat. In such a simulation, the relative positions of the Cα atoms were kept fixed (force constant 1 kcal/mol), in order to exclusively sample the arrangement of the side chains. Short-range electrostatic interactions were cutoff at 9 Å, a RESPA integrator was used with a time-step of 2 fs, and long-range electrostatics were computed every 6 fs. MD simulations were performed using the OPLS3 force field in Desmond 5.0 software (Schrödinger Release 2017-4: Desmond Molecular Dynamics System, D. E. Shaw Research, New York, NY, 2017) and run for 50 ns. Recording interval was set to 50 ps, allowing the collection of 1001 frames. The trajectory was clustered using the "Desmond trajectory clustering" tool in Maestro (Maestro-Desmond Interoperability Tools, Schrödinger) based on the RMSD of residues 152, 153, 156, 157, 158, 187, 196, 197, 198, 202, 203, 205, 206, and 209 (i.e., the residues composing the interested site). A hierarchical clustering was performed to obtain 10 clusters of the explored site. The centroid of each cluster was then selected as representative structure and subjected to in silico ligand-binding site prediction and druggability assessment by using the above-mentioned consensus methods involving DogSiteScorer and SiteMap analysis (Table S1).

*Virtual chemical library preparation.* The Asinex Gold & Platinum Library was downloaded from the Asinex webpage ($\sim$3.2 × 10⁵ commercially available compounds, www.asinex.com). A first round of ligand preparation was performed in LigPrep (Schrödinger Release 2017-4: LigPrep, Schrödinger). In this step, the different tautomeric forms for undefined chiral centers were created. By contrast, for

specified chirality, only the specified enantiomer was retained. Subsequently, the compounds were imported within SeeSAR (SeeSAR version 5.6, BioSolveIT GmbH), that assigned the proper geometry, the protonation state and the tautomeric form of the compounds using the ProToss method. A final library of $\sim$4.3 × 10⁵ docking clients was thus obtained.

*Identification of in silico hits through virtual screening.* The virtual screening workflow was developed by using the KNIME analytic platform and the BioSolveIT KNIME nodes. Specifically, the workflow was organized as follows: (i) the "Prepare Receptor with LeadIT" node was used for protein preparation and docking parameters definition in LeadIT (LeadIT version 2.2.0; BioSolveIT GmbH, www.biosolveit.de/LeadIT). The binding site was defined on the basis of the residues composing the identified druggable pocket (Fig. S2C). The residue protonation states, as well as the tautomeric forms, were automatically assessed in LeadIT using the ProToss method, that generates the most probable hydrogen positions on the basis of an optimal hydrogen bonding network using an empirical scoring function; (ii) the "Compute LeadIT Docking" node was selected to perform the docking simulations of the $\sim$4.3 × 10⁵ docking clients by using the FlexX algorithm[54]. Ten poses for each ligand were produced; (iii) the "Assess Affinity with HYDE in SeeSAR" node generated refined binding free energy (i.e., ΔG) and estimated HYDE affinity ($K_{iHYDE}$) for each ligand pose using the HYDE rescoring function;[55] (iv) for each ligand, the pose with the lowest $K_{iHYDE}$ was extracted. Only compounds with a predicted $K_{iHYDE}$ range below 5 μM were retained for the following steps; (v) the rescored poses were filtered based on physicochemical and ADME filters using the Optibrium models integrated in SeeSAR (Optibrium 2018). In particular the following filters were used: 2 ≤ LogP ≤ 5, where LogP is the calculated octanol/water partition coefficient; 1.7 ≤ LogD ≤ 5, where logD is the calculated octanol/water distribution coefficient; TPSA ≤ 90; where TPSA is the topological polar surface area; LogS ≥ 1, where a LogS corresponds to intrinsic aqueous solubility greater than 10 μM; LogS7.4 ≥ 1, where LogS7.4 is the intrinsic aqueous solubility at pH of 7.4; HIA = +, where HIA is the classification for human intestinal absorption (predicts a classification of "+" for compounds which are ≥ 30% absorbed and "−" for compounds which are < 30% absorbed); 300 ≤ MW ≤ 500, where MW is molecular weight; number of rotable bonds ≤ 3; PgP category = −, where PgP category is the classification of P-glycoprotein transport (the compound must belong to the "−" category to avoid the active efflux); number of hydrogen bond donors ≤ 3; number of stereocenters ≤ 1. In addition, molecules potentially acting as pan-assay interference compounds were discharged. This approach produced a list of 275 virtual hits, which were first submitted to a diversity-based selection. For each compound, a binary fingerprint was derived by means of the canvasFPgen utility provided by Schrödinger (Fingerprint type: MolPrint2D; precision:XP) (Schrödinger Release 2017-4: Canvas, Schrödinger). Using the created fingerprint, the 10 most different compounds (i.e., ASN 03578729, ASN 15755504, ASN 16356773, ASN 17325626, ASN 19380113, BAS 00312802, BAS 00340795, BAS 00382671, BAS 01058340, BAS 01849776) were extrapolated by applying the canvasLibOpt Schrodinger utility (Schrödinger Release 2017-4: Canvas, Schrödinger). In addition, visual inspection guided the selection of promising ligands based on the predicted binding mode and the interactions established with the identified binding pocket. In total, 30 molecules were selected, 8 from the diverse selection and 22 after visual inspection (Fig. S3 and Table S2). Indeed, even though 10 diverse compounds were originally chosen, BAS 00340795 was not in stock and ASN 03578729 was later replaced by its close analog ASN 05397475, selected by 3D visualization and endowed with a better predicted affinity.

## Chemical synthesis of SM875

*Reagents and instrumentation.* The reagents (Sigma Aldrich) and solvents (Merck) were used without purification. The reaction yields were not optimized and calculated after chromatographic purification. Thin layer chromatography (TLC) was carried out on Merck Kieselgel 60 PF254 with visualization by UV light. Microwave-assisted reactions were carried out using a mono-mode CEM Discover reactor in a sealed vessel. Preparative thin layer chromatography (PLC) on 20 × 20 cm Merck Kieselgel 60 F254 0.5 mm plates. High-performance liquid chromatography (HPLC) purification was performed by a Merck Hitachi L-6200 apparatus, equipped with a diode array detector Jasco UVIDEC 100 V and a LiChrospher reversed phase RP18 column, in isocratic conditions with eluent acetonitrile/water 1:1, flow 5 mL min⁻¹, detection at 254 nm. IR spectrum of the final product was recorded by using a FT-IR Tensor 27 Bruker spectrometer equipped with attenuated transmitter reflection device at 1 cm⁻¹ resolution in the region Δν 400 ÷ 1000 cm⁻¹. A thin solid layer was obtained by the evaporation of the chloroform solution in the sample. The instrument was purged with a constant dry nitrogen flow. Spectra processing was made using Opus software packaging. NMR spectra were recorded on a Bruker-Avance 400 spectrometer by using a 5 mm BBI probe ¹H at 400 MHz and ¹³C at 100 MHz in CDCl₃ relative to the solvent residual signals δH 7.25 and δC 77.00 ppm, J values in Hz. Structural assignments are confirmed by heteronuclear multiple bond correlation experiments. ESI-MS spectra were taken with a Bruker Esquire-LC mass spectrometer equipped with an electrospray ion source, by injecting the samples into the source from a methanol solution MS conditions: source temperature 300 °C, nebulizing gas N₂, 4 L min⁻¹, cone voltage 32 V, scan range m/z 100–900. LC-ESI-MS spectrum was acquired

using a C-18 Kinetex 5 μm column, eluting with acetonitrile/water 70:30, flow 1 mL min$^{-1}$ using ESI source as detector in positive ion mode. High-resolution ESI-MS measurement for the final product, including tandem MS2 fragmentation experiments, were obtained by direct infusion using an Orbitrap Fusion Tribrid mass spectrometer.

*Synthesis of SM875.* The target product SM875 was obtained according the synthetic strategy reported in Fig. S4. The sequence involves the preparation of the precursor 1-(4-bromophenyl)-1H-pyrazol-5-amine, which was used in a following three-component reaction with 4-hydroxy-3-methoxybenzaldehyde and Meldrum acid (2,2-dimethyl-1,3-dioxane-4,6-dione) according to a modified method[56]. The 1-(4-bromophenyl)-1H-pyrazol-5-amine was synthesized starting from 4-(bromophenyl) hydrazine that was obtained from the commercial hydrochloride by treatment with a saturated NaHCO$_3$ aqueous solution (50 mL), followed by dichloromethane extraction (50 mL ×3), treatment with anhydrous Na$_2$SO$_4$ and evaporation. To a magnetically stirred solution of 4-(bromophenyl)-hydrazine (100 mg, 0.53 mM, in 5 mL ethanol 5), ethyl-2-cyano-3-ethoxyacrylate (89.6 mg, 0.53 mM) was added and refluxed for 2 h. The reaction mixture was concentrated in vacuo, the residue was suspended in 1:1 methanol/2 M NaOH aqueous solution (5 mL) and refluxed for 1 h. After cooling, the mixture was neutralized with 1 M HCl aq. solution (5 mL) and concentrated in vacuo using a water bath at 40 °C. The residue was heated at 180 °C for 10 min, suspended in ethanol after cooling and stored overnight at 4 °C. The supernatant was recovered and concentrated to give a residue which was stirred in the presence of a NaHCO$_3$ solution (10 mL). Extraction with ethyl acetate (10 mL ×3), followed by the treatment with anhydrous Na$_2$SO$_4$ of the combined organic phases and concentration in vacuo gave the product (79 mg, 61%), which was used in the following three-component reaction. The successful synthesis of 1-(4-bromophenyl)-1H-pyrazol-5-amine was verified by $^1$H-NMR [δH 7.58 (d, J 8.7 Hz, H-3'and H-5'), 7.47 (d, J 8.7 Hz, H-2'and H-6'), 7.41 (s, H-3), 5.62 (s, H-4)] and ESI-MS (m/z 239.8 [M + H]$^+$). In the last reaction step, 1-(4-bromo-phenyl)-1H-pyrazol-5-amine (79 mg, 0.33 mM), 4-hydroxy-3-methoxybenzaldehyde (41 mg, 0.27 mM) and 2,2-dimethyl-1,3-dioxane-4,6-dione (46 mg, 0.32 mM) in ethanol (5 mL) were refluxed under stirring for 2.5 h, or alternatively by replacing the conventional heating with microwave irradiation at 110 °C for 1 h. The reaction mixture was then cooled to room temperature (RT) and dried in vacuo. The raw product was purified by silica preparative thin layer chromatography (PLC) eluting with n-hexane/ethyl acetate (1:1). The band collected at retention factor 0.4 was first used for structural characterization and then injected into preparative HPLC (RP18 column, acetonitrile/water 1:1, UV detection at 254 nm, flow 5 mL min$^{-1}$, retention time 4.5 min) to give the target product (for use on cell cultures) as a white powder after evaporation of the eluent: 34 mg, 25% with reflux in ethanol; a yield of ~25% is also obtained with microwave irradiation.

*Structural characterization of SM875.* NMR, HRESI-MS spectra together with LC-MS (UV, EIC, and MS) are reported in Fig. S5 and Table S3. Figure S5B, relative to HRESI-MS, reports the results of single measurements, whereas the average of 80 measures in positive ion mode is reported here: HRESI(+)MS: m/z 414.0443 ± 0.0013, [M + H]$^+$ (calculated for C$_{19}$H$_{17}$$^{79}$BrN$_3$O$_3$ = 414.0448); HRESI(+)MS/MS on the fragment ion of 414.0443: m/z 399.0205 ± 0.0019, [M + H-CH3]$^+$ (calculated for C$_{18}$H$_{14}$$^{79}$BrN$_3$O$_3$ = 399.0213); m/z 372.0335 ± 0.0019, [M + H-C$_2$H$_2$O]$^+$ (calculated for C$_{17}$H$_{15}$$^{79}$BrN$_3$O$_2$ = 372.0342); m/z 289.9918 ± 0.0017, [M + H-C$_7$H$_8$O$_2$]$^+$ (calculated for C$_{12}$H$_9$$^{79}$BrN$_3$O = 289.9929).

## Cell-based and biochemical assays

*Cell cultures and treatments.* Cell lines used in this paper have been cultured in Dulbecco's Minimal Essential Medium (Gibco, #11960-044), 10% heat-inactivated fetal bovine serum (Δ56-FBS), 50 U/mL penicillin, and 50 μg/mL streptomycin (Pen/Strep, Corning #20-002-Cl), non-essential amino acids (Gibco, #11140-035) and L-Glutamine (Gibco, #25030-024), unless specified differently. HEK293 and N2a cells were obtained from ATCC (ATCC CRL-1573 and CCL-131, respectively). We used a subclone (A23) of HEK293 stably expressing a mouse wild-type PrP or an EGFP-PrP construct, both already described and characterized previously[32]. L929 mouse fibroblasts and inducible RK13 cells were kindly provided by Ina Vorberg (DZNE, Bonn, Germany)[42] and Didier Villette (INRA, Toulouse, France)[38], respectively. Human cancer cell lines (H358, ZR-75, A549, H460, MCF7, H1299, SKBR3, and T47D), all belonging to the NCI collection of human cancer cell lines, were kindly provided by Valentina Bonetto (Mario Negri Institute, Milan, Italy). All the cell lines were originally authenticated by the provider (ATCC or NCI) and tested for possible mycoplasma contamination every two months. Cells were passaged in T25 flasks or 100 mm Petri dishes in media containing 200 μg/mL of Hygromycin or 500 μg/mL of G418 and split every 3–4 days. Every cell line employed in this study has not been passaged more than 20 times from the original stock. Compounds used in the experiments were resuspended at 30 or 50 mM in DMSO, and diluted to make a 1000× stock solution, which was then used for serial dilutions. A 1 μL aliquot of each compound dilution point was then added to cells plated in 1 mL of media with no selection antibiotics. For pulse experiments, inducible RK13 cells were seeded on 24-well plates at a confluence of 50%. After 24 h cells were treated with doxyciclin (0.01 mg/mL) or vehicle (0.1% DMSO), in the presence or absence of brefeldin 1 A (BREF 10 μM) or SM875 (10 μM). At the

end of each time-point (2, 4, 8 and 24 h) cells were washed with PBS and then lysed in lysis buffer. For chase experiments, RK13 cells were seeded on 24-well plates at a confluence of 30%. After 24 h cells were treated with doxyciclin (0.01 mg/mL) for 24 h. The medium containing doxycycline was then removed and cells kept in fresh medium for 4 h before adding SM875 (10 μM). After 5, 19, and 24 h of incubation cells wells were washed with PBS and lysed.

*Plasmids.* The EGFP-PrP construct contains a monomerized version of EGFP inserted after codon 34 of mouse PrP[57]. The identity of all constructs was confirmed by sequencing the entire coding region. All constructs were cloned into the pcDNA3.1(+)/hygro expression plasmid (Invitrogen). The Strep-FLAG pcDNA3.1 (+)/G418 NEGR-1 and the anti-NEGR-1 primary antibody were kindly provided by Giovanni Piccoli (University of Trento, Italy)[58]. All plasmids were transfected using Lipofectamine 2000 (Life Technologies), following manufacturer's instructions.

*Western blotting and antibodies.* Samples were lysed in lysis buffer (Tris 10 mM, pH 7.4, 0.5% NP-40, 0.5% TX-100, 150 mM NaCl plus complete ethylenediaminetetraacetic acid-free Protease Inhibitor Cocktail Tablets, Roche, #11697498001), diluted 2:1 in 4× Laemmli sample buffer (Bio-Rad) containing 100 mM Dithiothreitol (CAS No. 3483-12-3, Sigma Aldrich), boiled 8 min at 95 °C and loaded on sodium dodecyl sulfate polyacrylamide gel electrophoresis (SDS-PAGE), using 12% acrylamide pre-cast gels (Bio- Rad) and then transferred to polyvinylidene fluoride (PVDF) membranes (Thermo Fisher Scientific). Membranes were blocked for 20 min in 5% (w/v) non-fat dry milk in Tris-buffered saline containing 0.01% Tween-20 (TBS-T). Blots were probed with anti-PrP antibodies D18 (kindly provided by D. Burton, The Scripps Research Institute, La Jolla, CA) or 6D11 (1:5000) in BSA 3% in TBS-T overnight at 4 °C, anti-NEGR-1 antibody (1:5000, kindly provided by Giovanni Piccoli, University of Trento, Italy), anti-Thy-1 (1:5000, OX7, Abcam) or anti-LC3 (1:2000, ab51520, Abcam), all in 5% (w/v) non-fat dry milk. After incubation with primary antibodies, membranes were washed three times with TBS-T (10 min each), then probed with a 1:8000 dilution of horseradish conjugated goat anti-human (Jackson Immunoresearch) or anti-mouse (Santa Cruz) IgG for 1 h at RT. After two washes with TBS-T and one with Milli-Q water, signals were revealed using the ECL Prime western blotting Detection Kit (GE Healthcare), and visualized with a ChemiDoc XRS Touch Imaging System (Bio-Rad). In all the experiments, with the exception of PK-treated samples, the final quantification of proteins detected by primary antibodies were obtained by densitometric analysis of the western blots, normalizing each signal on the corresponding total protein lane (obtained by the enhanced tryptophan fluorescence technology of stain-free gels, BioRad).

*Dot blotting.* Samples were spotted on PVDF membrane in a 96-well dot blot system. A 96-well dot blot apparatus (Schleicher & Schuell) was set up with a 0.45-μm-pore-size PVDF membrane (Immobilon-P; Millipore), and each dot was rinsed with 500 μL of TBS. Under vacuum, cell lysates diluted 1:10 in TBS were added to the apparatus and rinsed with 500 μL of TBS. The membrane was then blocked with 5% milk-0.05% Tween-20 (Sigma) in TBS (TBS T-milk) for 30 min and probed with the anti-PrP antibody 6D11 (1:4000) followed by goat anti-mouse IgG (Pierce). Signals were revealed using enhanced chemiluminescence (Luminata, Bio-Rad) and visualized by a ChemiDoc XRS Touch Imaging System (Bio-Rad).

*Quantitative real-time PCR.* Following treatments, cells were harvested from 24-well plates and RNA was extracted using TRIzol (Invitrogen) or RNeasy Plus mini kit (Quiagen). An 800-ng aliquot per each sample was reverse transcribed using High Capacity cDNA Reverse Transcription kit (Applied Biosystems) according to the manufacturer's instructions. Quantitative RT-PCR was performed in a CFX96 Touch thermocycler (Bio-Rad) using PowerUp SYBR Green Master mix (Invitrogen) for 40 cycles amplification. Mouse PrP set 1 and set 2 were used to amplify endogenous and transgenic PrP, respectively (Table 1). Relative quantification was normalized to mouse or human HPRT (hypoxanthine–guanine phosphoribosyltransferase) as a housekeeping control.

*Immunofluorescence and high-content imaging.* Immunocytochemistry was performed on inducible RK13 cells treated for 2, 4, 8, or 24 h with doxycycline 0.01 (1×) or 0.1 (10×) mg/mL, in the presence or absence of SM875 (10 μM). Cells were seeded on CellCarrier-384 Ultra microplates (Perkin Elmer) at a concentration of 6000 cells/well, grown for ~24 h, to obtain a semi-confluent layer (60%) and treated with the compound. Cells were fixed for 12 min at RT by adding methanol-free paraformaldehyde (Thermo Fisher Scientific) to a final concentration of 4%. Wells were then washed three times with PBS, and permeabilized for 1 min with PBS containing a final concentration of 0.1% Triton X-100. Wells were washed three times with PBS and cells were incubated with blocking solution (FBS 2% in PBS) for 1 h at RT. The anti-PrP primary antibody (D18) was diluted in the blocking solution and added to the wells to a final concentration of 1:400. After three washes with PBS, the secondary antibody (Alexa 488-conjugated goat anti-human IgG diluted 1:500 in blocking solution) was incubated for 1 h at RT. Hoechst 33342 (Thermo Fisher Scientific) diluted in 0.5 mM PBS was then added after two additional washes. In another set of experiments, cells expressing EGFP-PrP were plated on CellCarrier-384 Ultra microplates (Perkin Elmer) at a

**Table 1 Primers employed to amplify endogenous and transgenic PrP.**

| Primer | Forward sequence 5′–3′ | Reverse sequence 5′–3′ |
|---|---|---|
| Mouse HPRT (endogenous) | TCAGACCGCTTTTTGCCGCGA | ATCGCTAATCACGACGCTGGGAC |
| Mouse PrP set 1 (endogenous) | GGACATCACCAAGACGAGGG | CGCCATGATGACTGATCCGA |
| Mouse PrP set 2 (transfected) | GCTGGCCCTCTTTGTGACTA | GTTCCACCCTCCAGGCTTTG |
| Human HPRT (endogenous) | CAGCCCTGGCGTCGTGATTAGTGA | TCACATCTCGAGCAAGACGTTCAGT |
| Human PrP (endogenous) | CCGAGGCAGAGCAGTCATTA | CCAGGTCACTCCATGTGGC |

concentration of 12.000 cells/well and grown for ~24 h, to obtain a semi-confluent layer (60%). SM875 was administered at a final concentration of 0.1, 0.3, 1, 3, or 10 μM, in two replicate wells. Vehicle (0.1% DMSO, volume equivalent) was used as a negative control. Cells were treated for 24 h and then fixed for 12 min at RT by adding methanol-free paraformaldehyde (Thermo Fisher Scientific) to a final concentration of 4%. Plates were then washed twice with PBS and counterstained with Hoechst 33342. The cell localization of EGFP-PrP and inducible PrP was monitored using an Operetta High-Content Imaging System (Perkin Elmer). Imaging was performed in a widefield mode using a ×20 High NA objective (0.75). Five fields were acquired in each well over two channels (380–445 Excitation-Emission for Hoechst and 475–525 for EGFP and Alexa 488). Image analysis was performed using the Harmony software version 4.1 (Perkin Elmer).

*Cell viability.* Cells were seeded on 24-well plates at ~60% confluence. Compounds at different concentrations or vehicle control (0.1% DMSO, volume equivalent) were added after 48 h, medium was replaced the second day, and then removed after a total of 48 h of treatment. Cells were incubated with 5 mg/mL of 3-(4,5-dimethylthiazol-2-yl)-2,5-diphenyltetrazolium bromide (MTT, Sigma Aldrich) in PBS for 15 min at 37 °C. After carefully removing MTT, cells were resuspended in 500 μL DMSO, and cell viability values obtained by a plate spectrophotometer (BioTek Instruments, VT, USA), measuring absorbance at 570 nm.

*Production of recombinant PrP.* RecHuPrP23-231 was expressed by competent *Escherichia coli* Rosetta (DE3) bacteria harboring pOPIN E expression vector[59]. Bacteria from a glycerolate maintained at −80 °C were grown in a 250 mL Erlenmeyer flask containing 50 mL of LB broth overnight. The culture was then transferred to two 2 L Erlenmeyer flasks containing each 500 mL of minimal medium supplemented with 3 g/L glucose, 1 g/L NH₄Cl, 1 M MgSO₄, 0.1 M CaCl₂, 10 mg/mL thiamine, and 10 mg/mL biotin. When the culture reached an OD600 of ~0.9–1.2 AU, Isopropyl β-D-1-thiogalactopyranoside was added to induce expression of PrP overnight under the same temperature and agitation conditions. Bacteria were then pelleted, lysed, inclusion bodies collected by centrifugation, and solubilized in 20 mM Tris-HCl, 0.5 M NaCl, 6 M Gnd/HCl, pH = 8. Although the protein does not contain a His-tag, purification of the protein was performed with a histidine affinity column (HisTrap FF crude 5 mL, GE Healthcare Amersham) taking advantage of the natural His present in the octapeptide repeat region of PrP. After elution with buffer containing 20 mM Tris-HCl, 0.5 M NaCl, 500 mM imidazole and 2 M guanidine-HCl, pH = 8, the quality and purity of protein batches was assessed by BlueSafe (NZYTech, Lisbon) staining after electrophoresis in SDS-PAGE gels. The protein was folded to the PrP native conformation by dialysis against 20 mM sodium acetate buffer, pH = 5. Aggregated material was removed by centrifugation. Correct folding was confirmed by CD and protein concentration, by measurement of absorbance at 280 nm. The protein was concentrated using Amicon centrifugal devices and the concentrated solution stored at −80 °C until used.

*Dynamic mass redistribution.* The EnSight Multimode Plate Reader (Perkin Elmer) was used to carry out DMR analyses. Immobilization of full-length (residues 23–230) or mouse N-terminally truncated (111–230) recombinant PrP (15 μL/well of a 2.5 μM PrP solution in 10 mM sodium acetate buffer, pH 5) on label-free microplates (EnSpire-LFB high sensitivity microplates, Perkin Elmer) was obtained by amine-coupling chemistry. The interaction between Fe³⁺-TMPyP, SM875 and SM940 diluted to different concentrations (0.03–100 μM, eight 1:3 serial dilutions) in assay buffer (10 mM NaHPO₃, pH 7.5, 2.4 mM KCl, 138 mM NaCl, 0.05% Tween-20) and PrP, was monitored after a 30 min incubation at RT. All the steps were executed by employing a Zephyr Compact Liquid Handling Workstation (Perkin Elmer). Data were obtained by normalizing each signal on the intra-well empty surface, and then by subtraction of the control wells. The Kaleido software (Perkin Elmer) was employed to acquire and process the data.

*Temperature-dependent detergent insolubility assay.* An 800 μM stock solution of freshly purified mouse recombinant PrP (residues 111–231) was diluted 1:10 in sodium acetate (10 mM NaAc, pH 7) to obtain 80 μM aliquots. To avoid precipitation of recombinant PrP, aliquots were flash frozen in liquid nitrogen, stored at −80 °C, and then kept on ice during their use. In order to carry out the assay, recombinant PrP was diluted to a final concentration of 0.5 μM in precipitation buffer (10 mM NaAc, 2% TX100, pH 7), split in eight identical aliquots, and

incubated for 1 h at different temperatures (25, 37, 45, and 55 °C), in the presence or absence of each molecule, or vehicle control (0.1% DMSO, volume equivalent). Each sample was then carefully loaded onto a double layer of sucrose (60% and 80%) prepared in precipitation buffer and deposited at the bottom of ultracentrifuge tubes. Samples were then subjected to ultracentrifugation at $100,000 \times g$ for 1 h at 4 °C. The obtained protein pellets were diluted in 2× LMSB and then analyzed by western blotting.

*Scanning electron microscopy.* SM875 (250 μM) was added to an 0.55 mL aliquot of recombinant PrP (250 μM in 20 mM PBS, pH = 6) and the solution was incubated for 1 h at 55 °C. Upon removal of supernatants, precipitates were collected by gentle resuspension, transferred to a vial and vigorously vortexed. A 10 μL aliquot was deposited on a glow-discharged gold-carbon grid. The grid was washed twice with d.i. water and stained for 1 min with a filtered, freshly prepared solution of 2% uranyl acetate. Samples were visualized using a ZEISS UltraPlus analytical FESEM with a grid stage set at 20 kV. For samples subjected to sonication, the same protocol was employed, with the exception of a short tip-sonication step (5 s, three times) before depositing the samples on the grid.

*Detection of prions in L929 fibroblasts.* L929 fibroblasts were grown in culturing medium and passaged 5–7 times after infection with a 0.5% homogenate of RML prion strain (corresponding to ~10⁵ lethal dose at 50%, derived from corresponding prion-infected mice; courtesy of Dr. Roberto Chiesa, Mario Negri Institute, Milan). In order to test the anti-prion effects of compounds, cells were seeded in 24-well plates (day 1) at ~60% confluence, with different concentrations of each molecule, or vehicle control (0.1% DMSO, volume equivalent). Medium containing fresh compounds or vehicle was replaced on day 2, and cells were split (1:2) on day 3, avoiding the use of trypsin by pipetting directly onto the well surface. Cells were collected on day 4 in PBS and centrifuged at 3500 rpm × 3 min. The resulting pellets were then rapidly stored at −80 °C. To evaluate prion loads, cell pellets were resuspended in 20 μL of lysis buffer (Tris 10 mM, pH 7.4, 0.5% NP-40, 0.5% TX-100, 150 mM NaCl) and incubated for 10 min at 37 °C with 2000 units/mL of DNase I (New England BioLabs). Half of the resulting sample was incubated with 10 μg/mL of PK (Sigma Aldrich) for 1 h at 37 °C, whereas the other half was incubated in the same conditions in the absence of PK. Both PK-treated and untreated samples were then mixed 1:2 with 4× Laemmli sample buffer (Bio-Rad) containing DTT, boiled for 8 min at 95 °C and ran by SDS-PAGE. The final quantification of PK-resistant PrP species was obtained by densitometric analysis of the western blots, normalizing each signal on the corresponding PK-untreated lane.

*Crystallization.* Recombinant mouse PrP (residues 111–230, 800 μM) was heated to 45 °C and slowly cooled to 20 °C using a thermal cycler, either in the presence or absence of SM875 (2 mM). Massive precipitation was observed in the presence of SM875. Pellet was removed by centrifugation and a 40% reduction in PrP concentration was estimated by UV absorbance at 280 nm. Showers of very thin needles appeared in 2–4 days (0.2 M (NH₄)₂SO₄, 10 mM CdCl₂, 9–14% LMW PEG smear, 9–14% MMW PEG smear, pH 6–8) irrespectively of whether the protein was incubated with SM875 or not. Data collection was performed at the Elettra synchrotron, XRD1 beamline (Trieste, Italy). Crystals were weakly and anisotropically diffracting to 3.7 Å in the best directions. Multiple diffraction patterns were present, as single needles were impossible to isolate. Orthorhombic crystal system and unit cell dimensions could be identified ($a = 36.1$ Å, $b = 51.8$ Å, $c = 55.9$ Å), which are extremely similar to those reported for apo human PrP in PDB 3HAK ($a = 32.5$ Å, $b = 49.1$ Å, $c = 56.9$ Å).

*Statistics and reproducibility.* All the data were collected and analyzed blindly by two different operators. Uncropped blot images of main figures are reported in Fig. S16. All source data underlying the graphs and charts presented in the main figures are available on Figshare (https://doi.org/10.6084/m9.figshare.13299209.v1). Statistical analyses, performed with the Prism software version 7.0 (GraphPad), included all the data points obtained, with the exception of experiments in which negative and/or positive controls did not give the expected outcome, which were discarded. No test for outliers was employed. The Kolmogorov–Smirnov normality test was applied (when possible, $n \geq 5$). Results were expressed as the mean ± standard errors, unless specified. In some case, the dose–response experiments were fitted with a four-parameter logistic (4PL) non-linear regression model, and

fitting was estimated by calculating the $R^2$. All the data were analyzed with the one-way analysis of variance test, including an assessment of the normality of data, and corrected by the Dunnet post-hoc test. Probability ($p$) values < 0.05 were considered as significant (*<0.05, **<0.01, ***<0.001).

**Reporting summary**. Further information on research design is available in the Nature Research Reporting Summary linked to this article.

## Data availability

All the data that support the findings of this study are available within the manuscript, supplementary files, or from the corresponding authors upon reasonable request. All source data underlying the graphs and charts presented in the main figures are available on Figshare (https://doi.org/10.6084/m9.figshare.13299209.v1).

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

## Acknowledgements

We thank Marco Sessa for designing the graphics shown in Fig. 8, Ina Vorberg (DZNE, Bonn, Germany) for mouse L929 fibroblasts, Didier Villette (INRA, Toulouse, France) for RK13 cells, Roberto Chiesa (Mario Negri Institute in Milan, Italy) for RML brain homogenates and Dennis Burton (The Scripps Research Institute, La Jolla, CA) for D18-expressing CHO cells. The authors are grateful to the staff of the XRD1 beamline at Elettra (Trieste, Italy) for on-site assistance, and acknowledge a CINECA award under the ISCRA initiative, for the availability of high-performance computing resources and support. L.To. is supported by Fondazione Caritro (Bando Post Doc 2017) and Kennedy's Disease Association (Research Grant 2018). Research of HCA was supported by the CJD Foundation, Inc. and the Alzheimer Forschung Initiative e.V. (AFI). J.R.R. was funded by a grant (BFU2017-86692-P) from the Spanish Ministry of Economy and Competitiveness, partially funded by FEDER funds. The work was also supported by grants from Fondazione Telethon and Provincia Autonoma di Trento to S.B. (TCP13007), from Fondazione Telethon to E.B. (TCP14009), and by a fellowship from Fondazione Telethon to G.S. S.B. and E.B. are Assistant Telethon Scientists at the Dulbecco Telethon Institute.

## Author contributions

Conceived and designed the computational analyses: G.S., A.A., A.I., S.O., A.B., L.T., L.To., M.P., M.L.B., P.F., E.B.; conceived and designed the experimental analyses: G.S., T.M., S.B., G.P., G.G., M.C., H.C.A., G.L., S.B., I.M., E.B.; performed the computational analyses: G.S., A.A., A.I., S.O., A.B., L.T., M.R., M.L.B., P.F.; performed the experimental analyses: G.S., T.M., S.B., P.B., M.L., V.B., G.M., N.L.L., L.C.F., Y.B.C., L.L., B.V., D.G., G.G., G.L., M.M.P., I.M.; analyzed the data: G.S., T.M., A.A., S.B., M.L., A.I., S.O., A.B., L.T., G.L., S.B., J.R.R., I.M., M.L.B., P.F., E.B.; contributed reagents/materials/analysis tools: none; wrote the paper: G.S. and E.B.; edited the paper: all authors.

## Competing interests

The authors declare the following competing interests: G.S., G.L., L.M.B., P.F., and E.B. are co-founders and shareholders of Sibylla Biotech SRL. The company exploits the PPI-FIT technology for drug discovery in a wide variety of human pathologies, with the exception of prion diseases.
