## [Peer Review File · Communications Biology]

Reviewers' comments:

Reviewer #1 (Remarks to the Author):

This is in my opinion an outstanding paper, easy to read, clear and very interesting. The idea at the basis of the manuscript is designing drugs capable of binding selectively a folding intermediate. This is a potentially revolutionary paradigm, which may open new perspectives for drug design, but which, I would have said before reading the paper, could be hindered by practical problems (the most important is how to deliver the drug where the protein folds). The paper provides compelling experimental evidence that the approach CAN work, at least in the case of the prion protein. I strongly appreciated the interdisciplinary approach, and the effort to provide evidence using very different techniques. Therefore, in my opinion the paper should be accepted for publication.

Minor revisions:

I suggest to discuss the relationship between this approach and the allosteric inhibition, in which one designs a drug which stabilizes a metastable conformation of a protein. Indeed, as explicitly stated in the manuscript, the intermediate state which is targeted in the design is also a metastable state, explored sporadically in a standard molecular dynamics slightly above room temperature. What is the competitive advantage of addressing a folding intermediate?

I suggest to discuss the possible general strategies for delivering a drug in the endoplasmatic reticulum, where most of the proteins fold.

Finally, I suggest discussing briefly if and how the design protocol could be modified in order to target chaperon-assisted folding

Reviewer #2 (Remarks to the Author):

The manuscript entitled "Pharmacological Inactivation of the Prion Protein by Targeting a Folding Intermediate" by Giovanni Spagnoli et al, described the new concept of drug discovery for protein misfolding diseases. They designed a novel drug discovery approach, Pharmacological Protein Inactivation by Folding Intermediate Targeting (PPI-FIT), in which folding intermediates of the cellular prion protein (PrP), can be key molecule for developing prion diseases, are targeted. They have built on computational simulation techniques to test a novel method for selectively reducing the level of target proteins. They identified small ligands for stabilizing a folding intermediate of a protein and showed the drugs could promote its degradation by the cellular quality control machinery, which could recognize such artificially stabilized intermediates as improperly folded species. In this manuscript, extensive characterization of one of these molecules, called SM875, provided strong experimental support for the hypothesis. The existence of intermediates has been suggested for long time but no one could proof that. The data also support the notion that the level of target proteins could be modulated by acting on their folding pathways, implying a previously unappreciated role for folding intermediates in the biological regulation of protein expression. They have conducted good amount of experiments carefully and the manuscript well described their experimental methods and results very well, and well organized to understand. It will be attracted for all researchers and readers who have interested in the protein-misfolding diseases and its treatment.

Reviewer #3 (Remarks to the Author):

The manuscript by Biasini and colleagues is a very nice contribution that I recommend for publication following some additional work aimed at increasing the relevance and applicability of their study to the broader biological sciences community. Their approach towards targeting a PrP intermediate for therapeutic purposes is carefully considered and aptly described. I commend the authors for the clarity in which they presented their findings and described their methods. This reviewer is not qualified to rigorously assess the computational aspects of their work, though they are well described and appear of good quality.

Major points:

1) The decrease in viability induced by the molecule is quite large at the concentrations utilized to attenuate PrP levels. The morphology of the cells is likely therefore to be compromised at higher concentrations of SM875. The fact that PrP mRNA and NEGR-1 levels were not changed by SM875 at these concentrations is encouraging, but I am concerned nonetheless that the level of cellular stress induced by the molecule alone over the course of the 24-hour treatment with the molecule could significantly impair cellular differentiation or membrane integrity, therein also indirectly changing the PrP levels. I recommend the authors augment their current description of Figure S7 in the text to thoroughly discuss any potential confounding variables. Simple brightfield images after treatment with SM875 or cell counts could help assess these points about cell number.

2) Given the authors' focus on the potential physiological relevance of this approach, the clarity of the SM875 mechanism of action could be augmented by including information regarding the secondary structure and morphology of the PrP aggregates that are formed in its presence. CD with recombinant PrP could determine if the PrP-SM875 species exhibit differential secondary structures, but FTIR might be better given that higher order protein aggregates can flatten and distort CD spectra. AFM or TEM would show if the morphology changes significantly, as the PrP-SM875 species could very well be off-pathway and this might help elucidate why they are easier to degrade. If the high-resolution imaging techniques are not readily available or straightforward, thioflavin-T measurements could assist in determining if the fibrillar mass fraction of PrP is changed by SM875 and would put the potential biological significance of this approach in better context, especially in light of the data suggesting that the molecule increases PrP aggregation in a temperature dependent fashion.

3) Considering the importance of oligomeric aggregates in numerous protein misfolding diseases and the potential generalizability of this approach to drug discovery against a wide range of human pathologies, a brief discussion of how PPI-FIT could be used to target neurotoxic oligomers (PrP, Amyloid-beta, alpha-synuclein, etc.) would be very helpful.

Minor points:

1) Saying that SM875 reduces PrP expression is somewhat unclear to this reviewer. The molecule did not change mRNA levels, but rather acted later to stimulate PrP degradation. Simply stating that it reduces total PrP levels would likely be clearer. The authors were clear in their description of the molecule and nothing is misleading, but the terminology could be more straightforward for general readers.

- 2) PrPSC is not defined explicitly in the introduction, nor is it used thereafter.
- 3) "in order to stabilize such _ intermediate" (add "an" where indicated)
- 4) "docking poses... is shown in Figure 2E" (change to are shown)
- 5) Regarding the cell lines used, the authors should indicate if any were mycoplasma tested or authenticated. I do not suggest they need to do so if they were not, but it would be good to include this information for clarity.
- 6) In all figure legends, it would be more clear to say what percentage of DMSO was used as the vehicle.
- 7) Figure 2 legend: are the independent replicates from different batches of cells (n=3 biologically independent experiments), or independent western blotting procedures of the same cells? I do not suggest they need to repeat any of these experiments, but making this point clear here and throughout the manuscript will be helpful. Details of how the cells were lysed and which antibodies were used for WB would be nice in the caption, too.
- 8) Figure 3 legend: do the dots in panel D represent independent experiments? Can the images in panel E be quantified and a histogram included to make the results more readily interpretable? Everywhere that p tests are described in the various figure legends, it would be helpful to also state the statistical test used.
- 9) "Bafilomycin A1 completely rescues BM875-induced..." (the rescue is not 100% at all concentrations, so I recommend rewording the sentence so it is in better agreement with the data).
- 10) Figure 5: The authors should show the experimental replicates as dots in panel C-ii, as they did elsewhere in the paper which was very good, and carry out the appropriate statistical tests to assist in the interpretation of these data.
- 11) "These results lay the groundwork for the chemical optimization..." (How so? Please elaborate to explain that this assay could be adapted to a high-throughput investigation of SM875 derivatives, if this is correct).
- 12) In the legend of figure 6, I see two periods after the title. It would also be helpful to label the blots with the bands that were quantified, as the authors did elsewhere (for example in Fig 5B). The caption says n=4 independent replicates, but I see 5 in some conditions. Please reword to match the data.
- 13) The concentration of PrP used in the partially denatured state experiments is really high (800 μ M). Could you please provide a short justification and or/discussion of the potential issues with this? Were there any solubility issues of the molecule alone at 2 mM in this buffer composition?
- 14) "The observed precipitation of partially-denatured...to expose hydrophobic residues normally buried" is very interesting and could be probed in bulk using ANS measurements to provide experimental evidence to this postulation. The authors could measure the ANS fluorescence, and

even possibly obtain a crude estimate of size by using turbidity of the exact same samples used in the ANS measurement, in a plate reader. The data presented and their claims are fair, so these measurements are not essential for this paper but could be useful for this or future studies.

15) Figure 7A: it would be more clear to label the lanes with the temperatures used rather than a ramping symbol.

16) “chimeras (PROTACs)59,.” (remove the extra comma)

17) The last sentence about Darwinian evolution seems speculative given their data.

18) In the “Cell Cultures and Treatments” methods section, the authors should include the concentration of Pen/Strep used in their experiments.

**UNIVERSITÀ
DI TRENTO**

Dipartimento di
Biologia Cellulare, Computazionale e Integrata

Trento, November 17, 2020

REBUTTAL LETTER

Reviewer #1:

This is in my opinion an outstanding paper, easy to read, clear and very interesting. The idea at the basis of the manuscript is designing drugs capable of binding selectively a folding intermediate. This is a potentially revolutionary paradigm, which may open new perspectives for drug design, but which, I would have said before reading the paper, could be hindered by practical problems (the most important is how to deliver the drug where the protein folds). The paper provides compelling experimental evidence that the approach CAN work, at least in the case of the prion protein. I strongly appreciated the interdisciplinary approach, and the effort to provide evidence using very different techniques. Therefore, in my opinion the paper should be accepted for publication.

We are grateful to the reviewer for the enthusiasm regarding our manuscript. We agree that the approach of targeting folding intermediates can be limited by the intrinsic nature of protein folding, generally occurring inside the cell. From this standpoint, PPI-FIT would be no different than classical pharmacological approaches directed against intracellular targets or from other degraders such as PROTACs, which aim at shuttling pharmacological targets to the intracellular degradation machinery. Following the reviewer's comment, we agree that any molecule designed to act by targeting a folding intermediate should possess the ability to cross lipid bilayers.

Minor revisions:

I suggest to discuss the relationship between this approach and the allosteric inhibition, in which one designs a drug which stabilizes a metastable conformation of a protein. Indeed, as explicitly stated in the manuscript, the intermediate state which is targeted in the design is also a metastable state, explored sporadically in a standard molecular dynamics slightly above room temperature. What is the competitive advantage of addressing a folding intermediate?

As correctly pointed out by the reviewer, the intermediate state could, in principle, be explored during folding or from the native conformation. In fact, this the case for PrP. While we cannot exclude that other proteins might behave differently, there are at least two main advantages of targeting a protein intermediate during the folding process. First, at room temperature, the intermediate state is predicted to be explored by almost every polypeptide during folding. Conversely, the rate of transitions from the native states to the folding intermediate is exponentially suppressed by the presence of an energy barrier. In addition, there is an intrinsic pharmacological advantage of targeting folding processes, which is the idea of activating the unfolding protein response against the target polypeptide and thus promote its degradation. As correctly suggested by the reviewer, we added a brief description of this point in the discussion section (page 13, lines 366-369).

I suggest to discuss the possible general strategies for delivering a drug in the endoplasmatic reticulum, where most of the proteins fold.

As anticipated above, molecules acting by binding folding intermediates should be able to cross lipid bilayers in order to reach their targets. It is well known that targeting proteins residing in organelles is a challenging task, and only limited information is available on this topic. For instance, structure-activity relationship analysis of known ER probes has provided some clues regarding the molecular features that direct small molecules to the ER [see for example: Louzoun-Zada et al. *Guiding Drugs to Target-Harboring Organelles: Stretching Drug-Delivery to a Higher Level of Resolution*. *Angewandte Chemie* 131.44 (2019): 15730-15740]. In particular, the amphipathic nature of a molecule can enhance the uptake into the ER because of the high concentration of zwitterionic lipid head-group in the ER membrane. Interestingly, in 2016, Mc Donalds and colleagues reported a series of amino-flavonoid able to be localized in the ER with high specificity [McDonald et al. *Fluorescent flavonoids for endoplasmic reticulum cell imaging*. *Journal of Materials Chemistry B* 4.48 (2016): 7902-7908]. Noteworthy, the authors also observed that more hydrophobic derivatives localized to the ER with higher selectivity. From an experimental perspective, this ability could be indirectly estimated by measuring the logarithm of partition ($\log P$) and distribution ($\log D$) coefficients, descriptors reflecting the relative concentrations of a compound in two immiscible solvents at equilibrium. Moreover, such a property could be evaluated in vitro using assays such as the parallel artificial membrane permeability assay (PAMPA). As suggested by the reviewer, we added a sentence specifying these concepts into the discussion section (page 13, lines 376-381).

Finally, I suggest discussing briefly if and how the design protocol could be modified in order to target chaperon-assisted folding

The reviewer raises an important point that likely affects the PPI-FIT approach. Theoretically, chaperone-assisted folding of proteins cannot be predicted by our algorithms. Moreover, druggable pockets appearing on the surface of folding intermediates could be masked by the chaperones, further limiting the ability to design effective drugs. Collectively, these considerations represent valuable points against the applicability of the PPI-FIT paradigm to proteins that fold by chaperone-assisted mechanisms. We stated this important limitation in the discussion section (page 13, lines 371-374).

Reviewer #2:

The manuscript entitled “Pharmacological Inactivation of the Prion Protein by Targeting a Folding Intermediate” by Giovanni Spagnolli et al, described the new concept of drug discovery for protein misfolding diseases. They designed a novel drug discovery approach, Pharmacological Protein Inactivation by Folding Intermediate Targeting (PPI-FIT), in which folding intermediates of the cellular prion protein (PrP), can be key molecule for developing prion diseases, are targeted. They have built on computational simulation techniques to test a novel method for selectively reducing the level of target proteins. They identified small

ligands for stabilizing a folding intermediate of a protein and showed the drugs could promote its degradation by the cellular quality control machinery, which could recognize such artificially stabilized intermediates as improperly folded species. In this manuscript, extensive characterization of one of these molecules, called SM875, provided strong experimental support for the hypothesis. The existence of intermediates has been suggested for long time but no one could proof that. The data also support the notion that the level of target proteins could be modulated by acting on their folding pathways, implying a previously unappreciated role for folding intermediates in the biological regulation of protein expression. They have conducted good amount of experiments carefully and the manuscript well described their experimental methods and results very well, and well organized to understand. It will be attracted for all researchers and readers who have interested in the protein-misfolding diseases and its treatment.

We are truly grateful to the reviewer for appreciating our work.

Reviewer #3:

The manuscript by Biasini and colleagues is a very nice contribution that I recommend for publication following some additional work aimed at increasing the relevance and applicability of their study to the broader biological sciences community. Their approach towards targeting a PrP intermediate for therapeutic purposes is carefully considered and aptly described. I commend the authors for the clarity in which they presented their findings and described their methods. This reviewer is not qualified to rigorously assess the computational aspects of their work, though they are well described and appear of good quality.

We sincerely thank the reviewer for the positive comments and useful suggestions.

Major points:

1) The decrease in viability induced by the molecule is quite large at the concentrations utilized to attenuate PrP levels. The morphology of the cells is likely therefore to be compromised at higher concentrations of SM875. The fact that PrP mRNA and NEGR-1 levels were not changed by SM875 at these concentrations is encouraging, but I am concerned nonetheless that the level of cellular stress induced by the molecule alone over the course of the 24-hour treatment with the molecule could significantly impair cellular differentiation or membrane integrity, therein also indirectly changing the PrP levels. I recommend the authors augment their current description of Figure S7 in the text to thoroughly discuss any potential confounding variables. Simple brightfield images after treatment with SM875 or cell counts could help assess these points about cell number.

The reviewer raised an important point regarding the possibility that at least part of the PrP lowering effects observed with SM875 could depend on the ability of the compound to alter membrane integrity. In fact, being PrP a membrane protein, a general reorganization of the plasma membrane could affect the levels of the proteins. We tried

to control such a possibility in a variety of ways. First, by checking that the expression of other non-relevant membrane proteins (NEGR-1 and Thy-1) was not affected by the compound. Moreover, the evaluation of cell viability upon treatment with the molecule in the different cell lines, and at the same incubation time and concentrations used in the other experiments, revealed cell-dependent cytotoxic effects (summarized in the legend of the Figure S7). Finally, to directly address the reviewer's suggestion, we added a new Figure (S8, page 36) showing the phase-contrast images of some of the cells employed in the study after treatment with SM875, which overall appear to correlate with cell viability assay. Collectively, we believe that these data indicate that at least the vast majority of the PrP lowering effects caused by SM875 are not related to the cytotoxicity of the molecule.

2) Given the authors' focus on the potential physiological relevance of this approach, the clarity of the SM875 mechanism of action could be augmented by including information regarding the secondary structure and morphology of the PrP aggregates that are formed in its presence. CD with recombinant PrP could determine if the PrP-SM875 species exhibit differential secondary structures, but FTIR might be better given that higher order protein aggregates can flatten and distort CD spectra. AFM or TEM would show if the morphology changes significantly, as the PrP-SM875 species could very well be off-pathway and this might help elucidate why they are easier to degrade. If the high-resolution imaging techniques are not readily available or straightforward, thioflavin-T measurements could assist in determining if the fibrillar mass fraction of PrP is changed by SM875 and would put the potential biological significance of this approach in better context, especially in light of the data suggesting that the molecule increases PrP aggregation in a temperature dependent fashion.

We thank the reviewer for this excellent suggestion. In order to characterize the structure of the PrP folding intermediate bound to SM875, we first tried to employ NMR. However, as occurred with the previous attempt to crystallize the complex, we observed substantial precipitation of PrP after adding SM875 upon mild thermal unfolding. Therefore, we sought to image such aggregated PrP species by using field emission scanning electron microscopy (FESEM). Protein precipitates withdrawn directly from NMR tubes were deposited on a glow-discharged gold-carbon grid, stained for 1 minute with 2% uranyl acetate, and visualized by FESEM using a ZEISS UltraPlus analytical Field Emission Scanning Electron Microscope (FESEM) with a grid stage set at 20 kV. Differently from amyloids of PrP prepared as previously described [Torrent et al., *Pressure reveals unique conformational features in prion protein fibril diversity*. *Sci Rep*. Feb 26;9(1):2802 (2019)], SM875 induced the formation of aggregates appearing as a multitude of dots of stained material of different sizes and shapes (~10 nm diameter) as well as much larger clumps. Sonication did not affect the morphology of these aggregates, leading only to the breakage of the larger clumps. Despite being at low resolution, these results confirm the reviewer's hypothesis that binding to SM875 induces the formation of PrP species that are off-pathway of the folding process. FESEM results and related discussion have now been added to the manuscript (new Figure S14, page 42; results discussed on page 11, lines 289-294).

3) Considering the importance of oligomeric aggregates in numerous protein misfolding diseases and the potential generalizability of this approach to drug discovery against a wide range of human pathologies, a brief discussion of how PPI-FIT could be used to target neurotoxic oligomers (PrP, Amyloid-beta, alpha-synuclein, etc.) would be very helpful.

This is a useful suggestion to highlight the possible applicability of the PPI-FIT. We have added the point in the discussion section (page 13, lines 374-376).

Minor points:

1) Saying that SM875 reduces PrP expression is somewhat unclear to this reviewer. The molecule did not change mRNA levels, but rather acted later to stimulate PrP degradation. Simply stating that it reduces total PrP levels would likely be clearer. The authors were clear in their description of the molecule and nothing is misleading, but the terminology could be more straightforward for general readers.

We agree with the reviewer that the use of the term “expression” was somehow misleading. Accordingly, we replace it with the terms “level”, “load” or “amount” throughout the text.

2) PrPSc is not defined explicitly in the introduction, nor is it used thereafter.

We introduced the definition as suggested (page 2, line 57).

3) “in order to stabilize such _ intermediate” (add “an” where indicated)

Added (page 2, line 86)

4) “docking poses... is shown in Figure 2E” (change to are shown)

Changed (page 4, line 154)

5) Regarding the cell lines used, the authors should indicate if any were mycoplasma tested or authenticated. I do not suggest they need to do so if they were not, but it would be good to include this information for clarity.

Our cell lines are passaged directly from the original stock authenticated by the different providers (ATCC or NCI) and routinely tested for mycoplasma (usually once every other month). We added this information into the Methods (page 25, lines 823-824).

6) In all figure legends, it would be more clear to say what percentage of DMSO was used as the vehicle.

DMSO was 0.1% in all samples/dilutions. We added this information in all the figure legends and Methods.

7) Figure 2 legend: are the independent replicates from different batches of cells (n=3 biologically independent experiments), or independent western blotting procedures of the same cells? I do not suggest they need to repeat any of these experiments, but making this point clear here and throughout the manuscript will be helpful. Details of how the cells were lysed and which antibodies were used for WB would be nice in the caption, too.

We added all the suggested information to the figure legend (page 5).

8) Figure 3 legend: do the dots in panel D represent independent experiments? Can the images in panel E be quantified and a histogram included to make the results more readily interpretable? Everywhere that p tests are described in the various figure legends, it would be helpful to also state the statistical test used.

We added all the suggested information to the figure legends. Unfortunately, images in Figure 3, panel E are not easy to quantify. An appropriate assessment would require segmentation of the membrane/intracellular space, which we tried to perform on these images, but realized they are at too low resolution for such an analysis (we also corrected a mistake in the figure legend, as the images have been taken with a fluorescence microscope and not with a high content imaging system as previously indicated, page 6). However, we would like to point out that a similar quantification is shown in Figure 5, panel C, where images have been acquired directly by using a high-content imaging system (Operetta, Perkin Elmer) and thus properly segmented and quantified.

9) “Bafilomycin A1 completely rescues BM875-induced...” (the rescue is not 100% at all concentrations, so I recommend rewording the sentence so it is in better agreement with the data).

We rephrased as follow: “Bafilomycin A1 largely rescues” (page 7, line 210).

10) Figure 5: The authors should show the experimental replicates as dots in panel C-ii, as they did elsewhere in the paper which was very good, and carry out the appropriate statistical tests to assist in the interpretation of these data.

We updated the figure as requested.

11) “These results lay the groundwork for the chemical optimization...” (How so? Please elaborate to explain that this assay could be adapted to a high-throughput investigation of SM875 derivatives, if this is correct).

After reading the section again, we found the sentence indicated by the reviewer somehow out of place. In addition, the concept of chemically optimize SM875 is already elaborated in the discussion section. Therefore, we removed the indicated sentence (page 9, lines 250-251).

12) In the legend of figure 6, I see two periods after the title. It would also be helpful to label the blots with the bands that were quantified, as the authors did elsewhere

(for example in Fig 5B). The caption says n=4 independent replicates, but I see 5 in some conditions. Please reword to match the data.

We corrected the typo. All the bands were quantified in the indicated figure. We updated the legend to explain it. We also changed the symbol to reflect that “at least” (\geq) 4 replicates were performed (page 9).

13) The concentration of PrP used in the partially denatured state experiments is really high (800 μ M). Could you please provide a short justification and or/discussion of the potential issues with this? Were there any solubility issues of the molecule alone at 2 mM in this buffer composition?

The very high concentrations of PrP and of the small molecule were required by the crystallization process. However, much lower concentrations were used in the detergent insolubility assay shown in Figure 7. Following the reviewer’s suggestion, we added a sentence to explain this point (page 10, lines 263-264 and 278-280).

14) “The observed precipitation of partially-denatured...to expose hydrophobic residues normally buried” is very interesting and could be probed in bulk using ANS measurements to provide experimental evidence to this postulation. The authors could measure the ANS fluorescence, and even possibly obtain a crude estimate of size by using turbidity of the exact same samples used in the ANS measurement, in a plate reader. The data presented and their claims are fair, so these measurements are not essential for this paper but could be useful for this or future studies.

The reviewer raised an interesting suggestion. As discussed in a previous point, the SM875-induced PrP aggregates were too large and insoluble to be measured by ANS fluorescence. We also tried by Size-Exclusion Chromatography, but again found that the aggregates didn’t even enter an S200 column (which we usually use to quantify oligomers of the amyloid β peptide, see for example, Fluharty et al. JBC 2013). After trying with FESEM (see a previous point), we may conclude that the task of characterizing these aggregates is not trivial, as it is the case for the vast majority of PrP assemblies, and we agree with the reviewer that it will certainly be a suitable subject for subsequent studies.

15) Figure 7A: it would be more clear to label the lanes with the temperatures used rather than a ramping symbol.

The panel has been updated as suggested (page 10).

16) “chimeras (PROTACs)59,.” (remove the extra comma)

Removed (page 13, line 382)

17) The last sentence about Darwinian evolution seems speculative given their data.

UNIVERSITÀ
DI TRENTO

Dipartimento di
Biologia Cellulare, Computazionale e Integrata

We believe the sentence may spark future interest in understanding the role of folding intermediates not only for pharmacological applications but also in physiological processes. However, we also agree with the reviewer that it may appear a bit speculative. Therefore, we rephrased the sentence to make it clear that it is mainly our interpretation of the results (page 14, lines 412-414).

18) In the “Cell Cultures and Treatments” methods section, the authors should include the concentration of Pen/Strep used in their experiments.

We added the requested information (page 24, lines 814-815).

REVIEWERS' COMMENTS:

Reviewer #3 (Remarks to the Author):

The authors have improved their manuscript by addressing the feedback of the reviewers, and I have no further concerns. I support the publication of their interesting study, with hope that they can leverage their approaches against additional protein misfolding diseases in the future.

**UNIVERSITÀ
DI TRENTO**

Dipartimento di
Biologia Cellulare, Computazionale e Integrata

Trento, November 30, 2020

REBUTTAL LETTER

Reviewer #3:

The authors have improved their manuscript by addressing the feedback of the reviewers, and I have no further concerns. I support the publication of their interesting study, with hope that they can leverage their approaches against additional protein misfolding diseases in the future.

We are grateful once again to the reviewer for the positive comments and constructive suggestions.